# High-throughput production of functional prototissues capable of producing NO for vasodilation

Xiangxiang Zhang[1], Chao Li[1], Fukai Liu[2], Wei Mu[1], Yongshuo Ren[1], Boyu Yang[1] & Xiaojun Han [1]✉

Bottom-up synthesis of prototissues helps us to understand the internal cellular communications in the natural tissues and their functions, as well as to improve or repair the damaged tissues. The existed prototissues are rarely used to improve the function of living tissues. We demonstrate a methodology to produce spatially programmable prototissues based on the magneto-Archimedes effect in a high-throughput manner. More than 2000 prototissues are produced once within 2 h. Two-component and three-component spatial coded prototissues are fabricated by varying the addition giant unilamellar vesicles order/number, and the magnetic field distributions. Two-step and three-step signal communications in the prototissues are realized using cascade enzyme reactions. More importantly, the two-component prototissues capable of producing nitric oxide cause vasodilation of rat blood vessels in the presence of glucose and hydroxyurea. The tension force decreases 2.59 g, meanwhile the blood vessel relaxation is of 31.2%. Our works pave the path to fabricate complicated programmable prototissues, and hold great potential in the biomedical field.

[1] State Key Laboratory of Urban Water Resource and Environment, School of Chemistry and Chemical Engineering, Harbin Institute of Technology, 92 West Da-Zhi Street, 150001 Harbin, China. [2] Animal Laboratory Center, The First Affiliated Hospital of Harbin Medical University, 23 You Zheng Street, 150001 Harbin, China. ✉email: hanxiaojun@hit.edu.cn

Natural tissues composed of multicellular components possess spatial hierarchical structures and collective behaviors triggered by intercommunications among cells[1,2]. Bottom-up fabrication of prototissues is beneficial to understanding the interaction mechanism among cells in the tissues, as well as holding great potential in the field of biomedical engineering[3–5].

The bottom-up assembled prototissues are roughly classified into two categories, i.e., randomly distributed and spatially well-defined structures. The randomly distributed prototissues were usually formed via noncovalent and covalent bonds. The prototissues composed of giant unilamellar vesicles (GUVs) were fabricated via electrostatic interactions[6], while the prototissues composed proteinsomes were formed via alkyne-azide cycloaddition reaction[7,8]. The spatially well-defined prototissues were produced via external forces, including optical[9], acoustic[10–12], and magnetic forces[13]. Optical tweezers were used to precisely build GUV colonies with various architectures of 2D (trigonal, square, and pentagonal) and 3D (tetrahedral, square-pyramidal, and three-layered pyramid) structures[9]. Acoustic fields were used to assemble various GUVs colony arrays, heterogeneous GUV/ natural cells, as well as dynamic colony arrays[11]. In our previous work, diverse spatial programmed GUVs prototissues were achieved based on the magneto-Archimedes effect[13]. Diamagnetic materials move to the weak magnetic field area in a non-homogeneous magnetic field. This phenomenon is called magneto-Archimedes effect. GUVs and living cells are diamagnetic materials, which were used as the building blocks for prototissues[13]. In addition, the 3D bioprinting technique was able to precisely code more building blocks into higher-order prototissues[14–16]. The economic high-throughput method is still on-demand to produce spatially coded prototissues.

Prototissues are not the simple aggregation of the protocells. The intercommunications among protocells and their functions are more important for their biomedical applications. The signal communications in prototissues have been achieved[4,5]. The two-step cascade enzyme reaction was often used for this purpose. The hydrogen peroxide (produced in one component by glucose and glucose oxidase) is transferred into other components containing horseradish peroxidase (HRP) to oxidize Amplex red into resorufin molecules in two-component prototissues[17,18]. The electric communication was realized in the 3D printing prototissues via light-triggered hemolysin expression[19,20]. The three-step cascade reaction among the components or more complicated network in the prototissues was rarely reported[21]. The functions of prototissues were demonstrated with thermal-responding reversible contractions and expansions[8], non-equilibrium biochemical sensing[22], and collective deformation[14]. There is no report to improve the real tissue function using prototissues.

Herein, a versatile method was demonstrated to fabricate high-throughput spatial programmable prototissues based on the magnetic Archimedes effect. Thousands of prototissues with spatially controllable structures were made in one go within 2 h. The signal communications were investigated both in two- and three-component spatially coded prototissues through enzyme cascade reactions. More importantly, we demonstrated the prototissues were able to cause blood vessel expansion due to the NO produced from prototissues triggered by glucose. The versatile methodology proposed in this work provides a path to fabricate complicated functional prototissues, which hold great potential in the biomedical field.

## Results and discussion
### Generation of prototissues using Magneto-Archimedes effect.
The spatial programmable prototissues were fabricated using the magneto-Archimedes effect in a homemade device (Fig. 1a and

Supplementary Fig. 1) containing a nickel mesh (NM) inside a petri-dish on the top of a circular permanent magnet. The magnetic potential energy $U(\mathbf{r})$ of a GUV with radius $R$ in the space magnetic field $\mathbf{H}(\mathbf{r})$ is given by formula ($\mathbf{1}$)[23].

$$U(\mathbf{r}) = -2\pi R^3 \mu_0 \frac{\chi_v - \chi_s}{\chi_v + 2\chi_s + 3}|\mathbf{H}(\mathbf{r})|^2 \qquad (1)$$

where $\chi_v$, $\chi_s$, and $\mu_0$ are the susceptibility of the GUV, the para-magnetic solution, and the permeability of the free space, respectively. Here, the magnetic potential energy $U(\mathbf{r})$ is positive because $\chi_v$ is less than $\chi_s$. Magnetic force drives GUVs to the regions with lower magnetic field strength, according to the energy minimum principle. Therefore, the spatial structure of the prototissues can be predicted according to the distribution of the magnetic field. The NM exhibits a strong magnetic response and causes a gradient magnetic field inside NM (Fig. 1b, bottom layer) and 140 μm above NM (Fig. 1b, top layer), where the dark blue areas indicate weak magnetic field regions. The weak magnetic field regions appear close to the wires in each grid on the NM surface (Fig. 1b, bottom layer), but in the center inside each grid 140 μm above NM surface (Fig. 1b, top layer). The NM was made with nickel wires (d = 60 μm). Each grid was 210 μm in the square.

As expected, the GUVs were firstly assembled in the weak magnetic field regions (dark blue areas in Fig. 1b, bottom layer) inside each grid (Fig. 1c I). With the number of added GUVs increasing, GUVs gradually filled the space inside each grid of the NM to generate 60 μm-thick bottom layer (Fig. 1c II) and further protruded to form 140 μm-thick brick-shape top layer above the GUVs aggregations inside each grid (Fig. 1c III, IV), which were consisted with simulated field distribution (dark blue areas in Fig. 1b, top layer). The concentrations of the GUVs for fabricating the prototissues in Fig. 1c were $3 \times 10^5$/mL for I, $5 \times 10^5$/mL for II, $8 \times 10^5$/mL for III, and $1.2 \times 10^6$/mL for IV, respectively. The 3D reconstructed confocal fluorescence images clearly showed the two-layer structures of GUV aggregates, containing a brick-shaped top layer (Fig. 1d, left image) and a square bottom layer (Fig. 1d, right image). The differential interference contrast (DIC) microscope image further demonstrated the two-layer structure arrays in the NM (Fig. 1e), where the brighter regions indicated the protruded brick-shape layer marked by a yellow dashed box (Fig. 1e, right image). We named this structure the protruded structure.

This prototissues fabrication method is easily scaled up to generate more than 2000 prototissues one time by using a bigger NM (Supplementary Fig. 2). The diversity of this method is also demonstrated by producing the various sizes of prototissues by using various NM with different lengths of the grid (Supplementary Fig. 3). In contrast, in the absence of a magnetic field, GUVs sank on the substrate with random distribution (Supplementary Fig. 4). When the NM was placed on the edge region of the top surface of the circular magnet (Fig. 1f), the magnetic field showed a distinct gradient distribution (Fig. 1g). In this condition, the size of the prototissues in the NM decreased gradually from the center to the edge regions of the circular magnet (Fig. 1h). The red dashed section line conformed to the gradient distribution of the prototissues in Fig. 1h. More importantly, this method was used to fabricate spatial programable multicomponent prototissues by varying the spatial magnetic field and the addition order of different GUVs (or cells).

### Diverse multicomponent prototissues.
The biological tissue functions depend on diverse tissue structures and heterogeneous compositions[4]. Herein, rGUVs (red fluorescence), gGUVs (green fluorescence), and non-labeled GUVs were used as building blocks to generate multicomponent prototissues with diverse

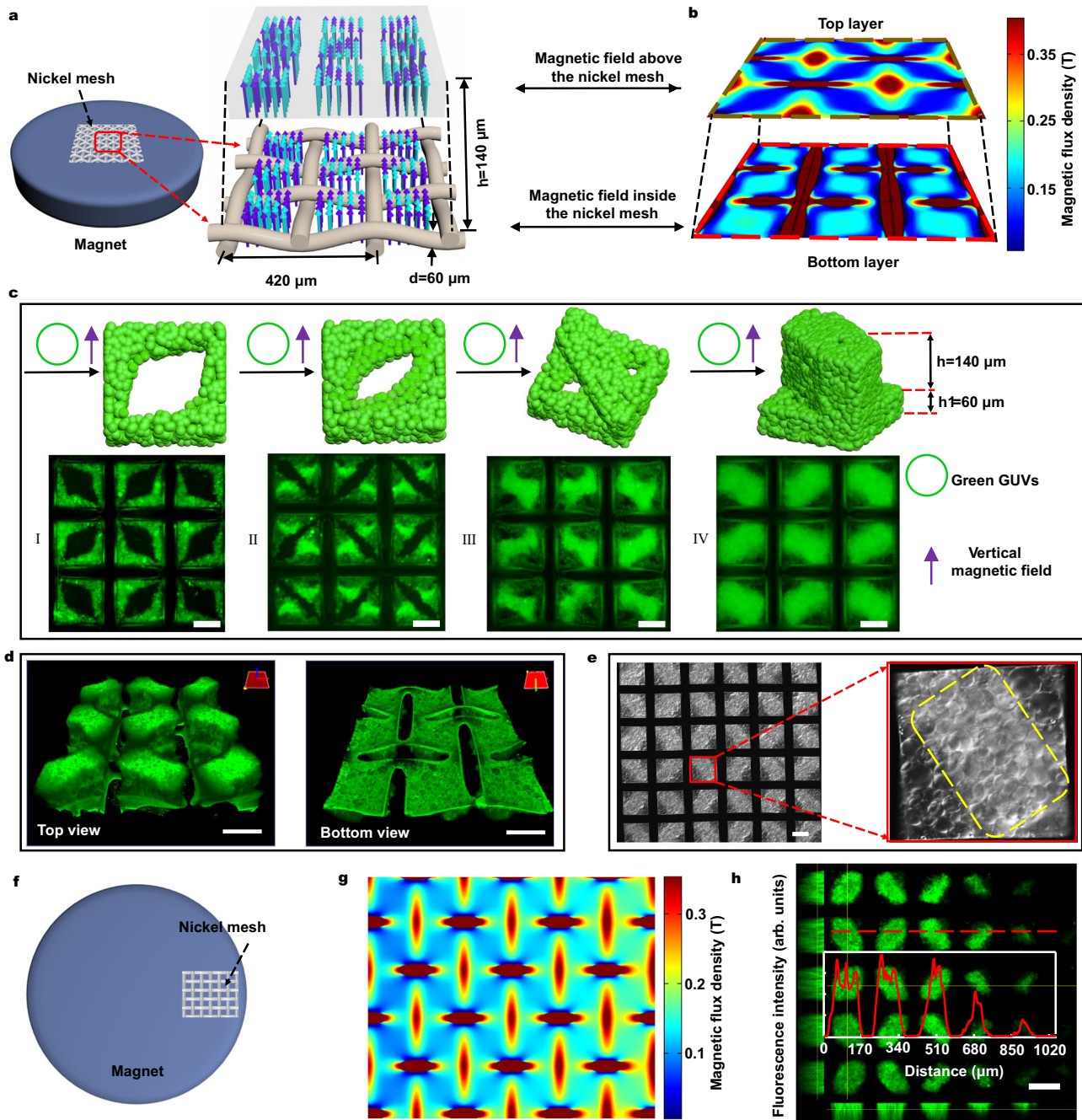

**Fig. 1 Generation of prototissues using Magneto-Archimedes effect. a** Schematic of the device with a woven nickel mesh (NM) at the center of a circular magnet. Cyan and blue arrows represented the strong and weak magnetic field, respectively. **b** Simulation results of the magnetic field of the bottom layer and top layer, which indicated the distribution of the magnetic field inside and above the NM, respectively. Dark blue areas indicated the weak magnetic field regions. **c** Schematic and fluorescence images (from at least five independent samples) of the prototissues assembled in a vertical magnetic field. With the number of added green giant unilamellar vesicles (gGUVs) increasing, the prototissues changed from single layer (I, II) to double layers (III, IV). The vertical purple arrow indicated the vertical magnetic field. **d** 3D reconstructed confocal fluorescence images of the prototissue array with protruded structure from top (left) and bottom (right) views. **e** A differential interference contrast (DIC) image (from at least three independent samples) of the 3D prototissue array. The lighter region in the image indicated the protruding top layer, which was marked by the yellow dashed box in the right image. **f** Schematic of the NM at the edge region of the circular magnet. **g** The simulation result of the magnetic field of the NM in **f**. **h** A fluorescence image (from three independent samples) of gGUVs prototissues in the magnetic field (**g**). Red line in the inset image corresponded to dashed line intensity analysis. The scale bars were 100 μm.

structures. Spatial programmable prototissues can be obtained by tuning the addition order/number of different GUVs and the distribution of magnetic fields. With a vertical field, the prototissues with rGUVs inside gGUVs colonies (Fig. 2b) were fabricated by adding gGUVs ($3 \times 10^5$/mL) and rGUVs ($2 \times 10^5$/mL)

successively. With a vertical field for gGUVs ($3 \times 10^5$/mL) assembly, while with an inclined field for rGUVs ($1 \times 10^5$/mL), the rGUVs was controlled to locate at the edge of half green eye structures (Fig. 2c). With both inclined fields for successive addition of gGUVs ($1 \times 10^5$/mL) and rGUVs ($1 \times 10^5$/mL),

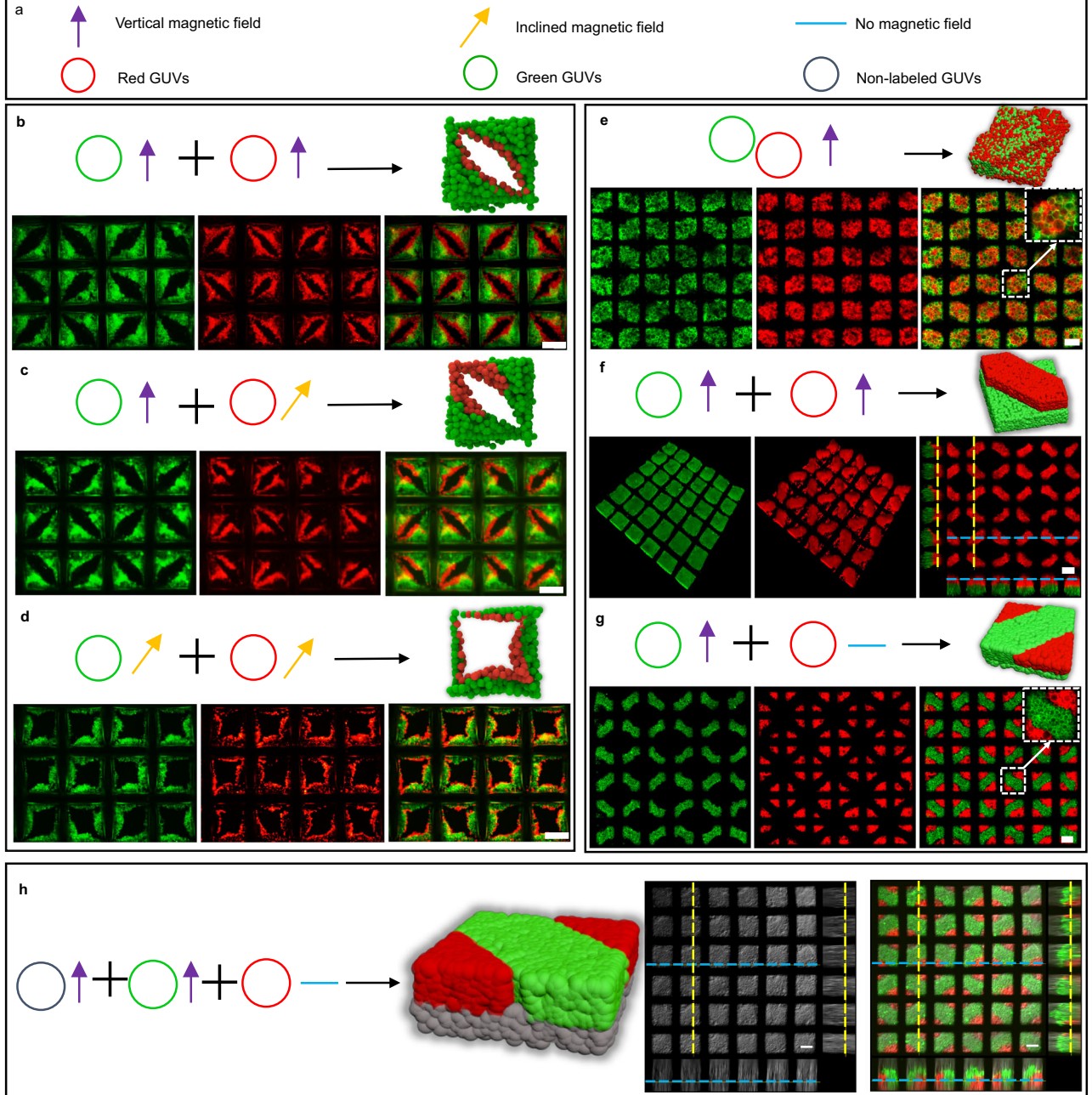

**Fig. 2 Diverse multicomponent prototissues. a** Purple arrow, yellow arrow, and cyan line indicated the vertical, inclined, and no magnetic field, respectively. Red, green, and gray rings indicated the red, green, and non-labeled GUVs, respectively. **b** Schematic and fluorescence images of a GUVs prototissue of eye structures with rGUVs inside gGUVs under a vertical magnetic field with the addition of gGUVs ($3 \times 10^5$/mL) and rGUVs ($2 \times 10^5$/mL) successively. **c** Schematic and fluorescence images of a GUVs prototissues of modified eye structures by trapping successively gGUVs ($3 \times 10^5$/mL) under a vertical magnetic field and rGUVs ($1 \times 10^5$/mL) under an inclined magnetic field. **d** Schematic and fluorescence images of prototissues by trapping successively gGUVs ($1 \times 10^5$/mL) and rGUVs ($1 \times 10^5$/mL) under an inclined magnetic field. **e** Schematic and fluorescence images of binary prototissues of protruded structures by trapping the mixture of gGUVs ($6 \times 10^5$/mL) and rGUVs ($6 \times 10^5$/mL) under a vertical magnetic field. **f** Schematic and fluorescence images of prototissues of protruded structures with gGUVs at the bottom and rGUVs at the top by trapping successively gGUVs ($6 \times 10^5$/mL) and rGUVs ($4 \times 10^5$/mL) under vertical magnetic field. **g** Schematic and fluorescence images of prototissues with two-layered structures by trapping successively gGUVs ($1.2 \times 10^6$/mL) under a vertical magnetic field and rGUVs ($4 \times 10^5$/mL) in the absence of magnetic field. **h** Schematic and fluorescence images of prototissues composed of three components by trapping successively non-labeled GUVs ($6 \times 10^5$/mL) and gGUVs ($6 \times 10^5$/mL) under vertical magnetic field to form protruded structrues, and subsequently rGUVs ($2 \times 10^5$/mL) in the absence of magnetic field. All the prototissues were assembled on the top of a circular magnet with 0.3 T magnetic flux density. After one type of GUVs were trapped, the time intervals were 1 h before adding another type of GUVs. The representative fluorescence images were from at least three independent samples. The scale bars were 100 μm.

the prototissues inside each grid were shown in Fig. 2d and Supplementary Fig. 5. Thus, we demonstrated that the spatial positions of components were controlled by adjusting the magnetic field and the addition order/number of components.

With the addition of more vesicles, the two-layered two-component prototissues were fabricated. With a vertical field, the structures similar to Fig. 1c IV but composed of rGUVs ($6 \times 10^5$/mL) and gGUVs ($6 \times 10^5$/mL) were obtained with the addition of the mixture of rGUVs and gGUVs (Fig. 2e). The green and red GUVs were evenly mixed in the prototissues. With a vertical field, the protruded structures (Fig. 2f) with the green bottom layer and red top layer were obtained by adding gGUVs ($6 \times 10^5$/mL) and rGUVs ($4 \times 10^5$/mL) successively. The projected images in Fig. 2f (right image) clearly showed the protruded two-layered structures. With a vertical field, the protruded structures were formed with gGUVs ($1.2 \times 10^6$/mL) first. Subsequently with the addition of rGUVs ($4 \times 10^5$/mL) with no magnetic field, the rGUVs filled in the rest regions of the top layer to form the structures as shown in Fig. 2g.

More diversely, the three-components spatial coded prototissues (Fig. 2h) were obtained with the addition of non-labeled GUVs ($6 \times 10^5$/mL) in a vertical field first, green GUVs in a vertical field second ($6 \times 10^5$/mL), and red GUVs with no magnetic field finally ($2 \times 10^5$/mL). The projected images in Fig. 2h (right image) clearly showed the wanted two-layered structures. The ratio of different GUVs can be adjusted freely to form similar structures (Supplementary Fig. 6). The three-component GUVs prototissues were stable on the nickel mesh when it was inverted for 120 min (Supplementary Fig. 7a). The three-component prototissues were maintained intact for 18 days (Supplementary Fig. 7b). Based on the principle demonstrated above, more complicated programmed spatial coded prototissues can be obtained by varying the magnetic field and the addition order/number of GUVs, which makes great sense in building complicated artificial or living tissues.

**Behaviors of the three-component prototissues under different osmotic stress conditions.** For the prototissues composed of GUVs encapsulating 300 mM sucrose, a hypotonic condition ($\Delta\Pi = 743.3$ kPa) was created by replacing the solution with pure water. The areas in yellow boxes (at hypotonic condition, Supplementary Fig. 8a2) became larger than those in cyan boxes (at isotonic condition, Supplementary Fig. 8a1), which indicated the expansion of green GUVs populations (assembled with magnetic field). Meanwhile, the tissue volume increased (Supplementary Fig. 8a4, a5), because the height of the projected image at hypotonic condition (yellow box in Supplementary Fig. 8a5) was larger than that at isotonic condition (cyan box in Supplementary Fig. 8a4) at the same cross-sections. By overlapping the cyan boxes and yellow boxes (Supplementary Fig. 8a3), the expansion percentage of GUVs populations was 16.41% (Supplementary Fig. 8d, red box) from $(A_a - A_b)/A_b \times 100\%$, where $A_a$ and $A_b$ were the average areas under hypotonic and isotonic conditions respectively. Similarly, the prototissue volume increased by 35.28% according to the heights of cyan and yellow boxes in Supplementary Fig. 8a6. The areas of red GUVs populations decreased by 26.99% (Supplementary Fig. 8e, red box) due to the confinement of the nickel mesh grids. At isotonic conditions, the density of the green GUVs assembled with a magnetic field was $10571 \pm 1378$/mm² , which was 1.66 times that of red GUVs assembled without a magnetic field ($6353 \pm 1066$/mm²) (Supplementary Fig. 8g). At the hypotonic conditions, the expanded green GUVs populations ($7628 \pm 673$/mm²) pushed the red GUVs into close-packed structures ($6904 \pm 1022$/mm²) (Supplementary Fig. 8g). On the contrary, the area of the green GUVs populations shrank by 7.41% (Supplementary Fig. 8b1, b2, b3,

and 8d green box) and 18.40% (Supplementary Fig. 8c1, c2, c3, and 8d purple box) under hypertonic conditions when the solution was replaced with 600 mM ($\Delta\Pi = -743.3$ kPa) and 900 mM ($\Delta\Pi = -1486.6$ kPa) glucose solution, respectively. The red GUVs populations with relative loose structure filled the space caused by the shrinking green GUVs populations, which resulted in the red GUVs populations areas increasing by 13.40% and 61.57%, respectively (Supplementary Fig. 8e, green and purple boxes). The prototissues volume decreased by 26.86% and 31.96% under the hypertonic conditions of 600 mM and 900 mM glucose solution, respectively (Supplementary Fig. 8f, green and purple boxes). The three-component prototissues were stable under hypertonic and hypotonic conditions. They exhibited collective expansion and contraction behaviors.

**Signal communication between two-component spatial coded prototissues.** In living systems, tissues are composed of heterogeneous cell populations with spatial distributions. Their functions are controlled by the signal communications among cells. After demonstrating the spatially coded prototissues with multicomponents, the signal communications among them were investigated. The prototissues similar to those in Fig. 2g were prepared using green GOx-gGUVs (with melittin pores in the lipid bilayer membrane, and glucose oxidases inside GUVs) and non-labeled HRP-GUVs (with horseradish peroxidases inside) (Fig. 3). The GOx-gGUVs formed the protruded structure at a vertical magnetic field, while the HRP-GUVs occupied the rest top layer regions with no magnetic field, as shown in the left image of Fig. 3b. The green fluorescent image (left image in Fig. 3c) and the merged image of green and bright-field image (right image in Fig. 3c) of the produced prototissues confirmed the designed structures. The signal communication between these two types GUVs was demonstrated using the cascade enzyme reactions shown in Fig. 3a. To initiate the reactions, glucose molecules (30 mM) and Amplex Red (0.05 μM) were added to the bath solution simultaneously. The added glucose molecules entered into GOx-gGUVs via melittin pores in the bilayer membrane, subsequently to be oxidized by GOx to produce $H_2O_2$. $H_2O_2$ diffused into nearby non-labeled HRP-GUVs to oxidize Amplex Red to generate red fluorescent resorufin catalyzed by HRP. As the reactants entered into the GUVs continuously, the red fluorescence gradually became stronger and reached the equilibrium in the non-labeled GUVs regions at about 20 min (Fig. 3d, e), which was three times faster than that from the floating GOx-GUVs and HRP-GUVs at the same experimental conditions (Supplementary Fig. 9). It implied the close packing of GUVs in the prototissues promoted chemical communications. Using GOx-free gGUVs to replace the GOx-gGUVs in the structures, no red fluorescent resorufin was observed (Fig. 3e, black line). Thus, we demonstrated the signal communication between two-component spatial coded prototissues.

**Signal communication in three-component spatial coded prototissues.** Furthermore, the signal communications among three-component prototissues on the NM were demonstrated as schematically shown in Fig. 4a, b. The protocol for fabricating the prototissues in Fig. 2h was used to form the prototissues composed of three components. The bottom layer and protruded layer were composed of non-labeled GOx-GUVs and C6 glioma cells, respectively. The rest regions of the top layer were occupied with Arginine-rGUVs (red TR-DHPE-labeled GUVs containing 20 mM L-Arginine). To initiate the ternary signal communications, the added glucose (30 mM) entered the non-labeled GOx-GUVs from the melittin pores to generate $H_2O_2$ by GOx (Fig. 4b, left image),

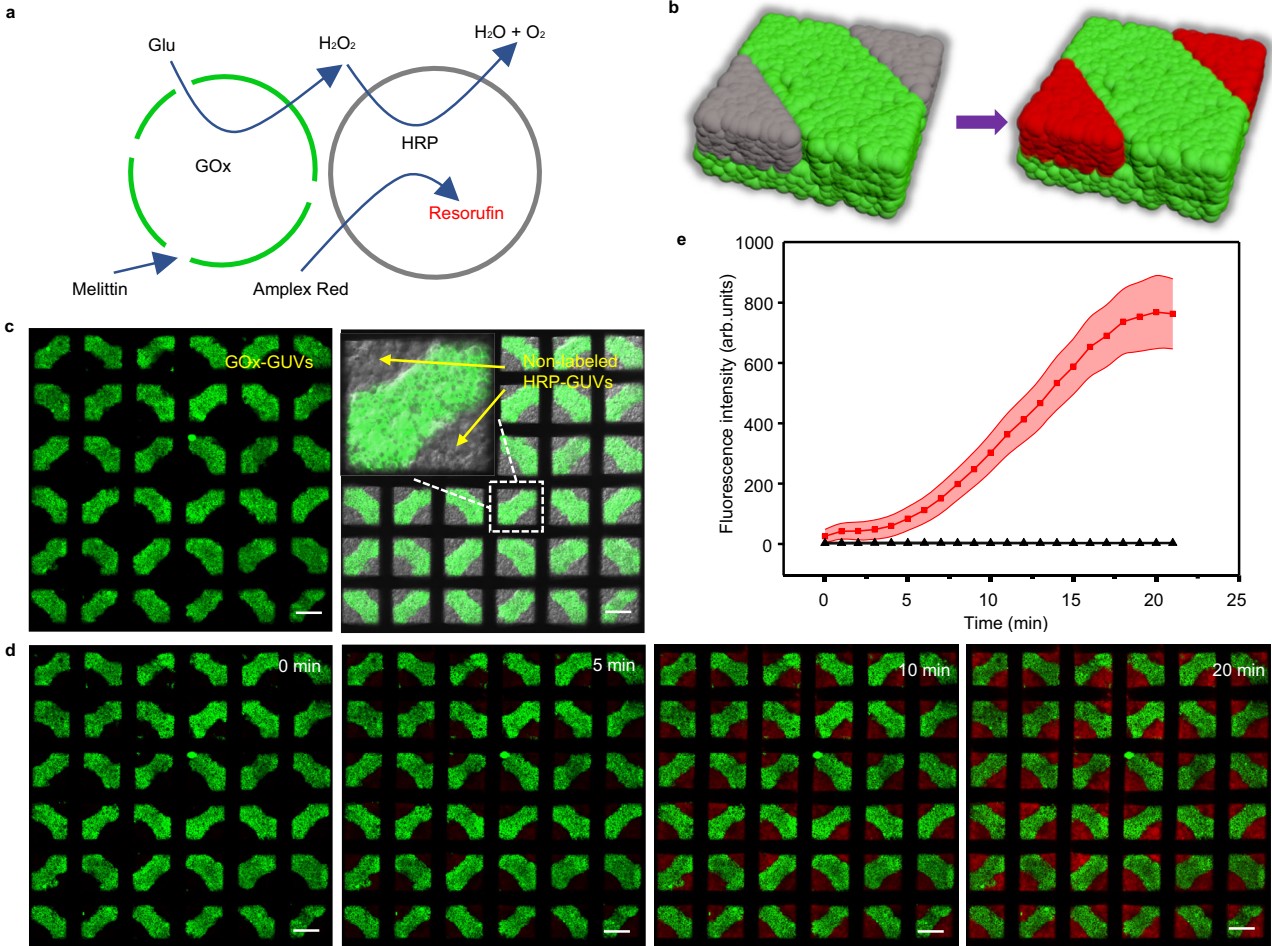

**Fig. 3 Signal communication between two-component spatial coded prototissues. a, b** Schematic illustration of signal communication in the prototissues composed of GOx-gGUVs with melittin pores and non-labeled HRP-GUVs. GOx and HRP indicated glucose oxidase and horseradish peroxidase, respectively. **c** Fluorescence image of the GOx-gGUVs populations (left), the merged image of the fluorescence and bright-field images of the GOx-gGUVs and non-labeled HRP-GUVs populations from three independent samples. **d** Confocal fluorescence images (from three independent samples) of the prototissues as a function of time after the addition of 30 mM glucose and 0.05 μM Amplex Red. **e** The mean fluorescence intensities in the HRP-GUVs regions in the images in **d** (red line), and the control samples using gGUVs to replace GOx-gGUVs (black line). The error bars were the standard deviation (SD, n = 5). The scale bars were 100 μm.

which entered into the Arginine-rGUVs and reacted with L-Arginine to release nitric oxide (NO)[24,25]. NO then entered the living C6 glioma cell populations and interacted with NO fluorescent probe of DAF-FM dyes in cells, which generated highly fluorescent triazole derivatives (DAF-T) (Fig. 4b, right image)[26]. The experimental results (Fig. 4c) confirmed the abovementioned signal communication pathway. The protruded C6 cell regions exhibited weak green fluorescence at 0 h, but gradually stronger green fluorescence as a function of time (Fig. 4c). The corresponding fluorescence intensity curve was shown in Fig. 4d (green curve). There was almost no increase in the fluorescence signal when GOx-GUVs were replaced by GOx-free GUVs (Fig. 4d, red curve). However, when L-Arginine was absent in rGUVs, the fluorescence intensity showed a low-level increase, which was probably due to the inherent L-Arginine in cells interacting with $H_2O_2$ (Fig. 4d, blue curve). A confocal fluorescence image with projected images of prototissues composed of three components showed the positions of non-labeled GOx-GUVs populations (the gray areas) at the bottom layer, Arginine-rGUVs populations (the red areas) at the edge of the top layer, and cell populations (the green areas) at the top layer (Fig. 4e and Supplementary Fig. 10). The glioma cells in the prototissues showed a similar death rate curve to the free cells as a function of time in the PBS solution,

which confirmed that the GUVs prototissue did not affect cell viability (Supplementary Fig. 11). The glioma cells in the hybrid prototissues in the PBS solution survived longer than the cells in the prototissues composed of GUVs (capable of producing $H_2O_2$ in the presence of glucose) and cells in the glucose solution[13], which was due to the toxicity of $H_2O_2$ generated by GUVs.

**Prototissues capable of producing NO.** The functions of prepared prototissues triggered by the internal signal communications were investigated in the following contents. Given NO served commonly as a second messenger involved in many physiological functions, such as inducing the relaxation of blood vessels[27,28], the prototissues were detached from the NM grids by shaking for the potential biomedical application. They still kept their morphology because of the hemi-fusion among gGUVs by incubating in 100 mM $CaCl_2$ for ten minutes before they were detached from NM grids (Fig. 5a). The detached prototissues survived for 8 days (Supplementary Fig. 12a). There were almost no GUVs dissociating from the prototissues by measuring the fluorescence of the solution in the petri-dish within 8 days (Supplementary Fig. 12b, c, and d). The prototissues were disassembled into dispersed individual GUVs and amorphous GUVs blocks when they were detached from nickel mesh without $CaCl_2$

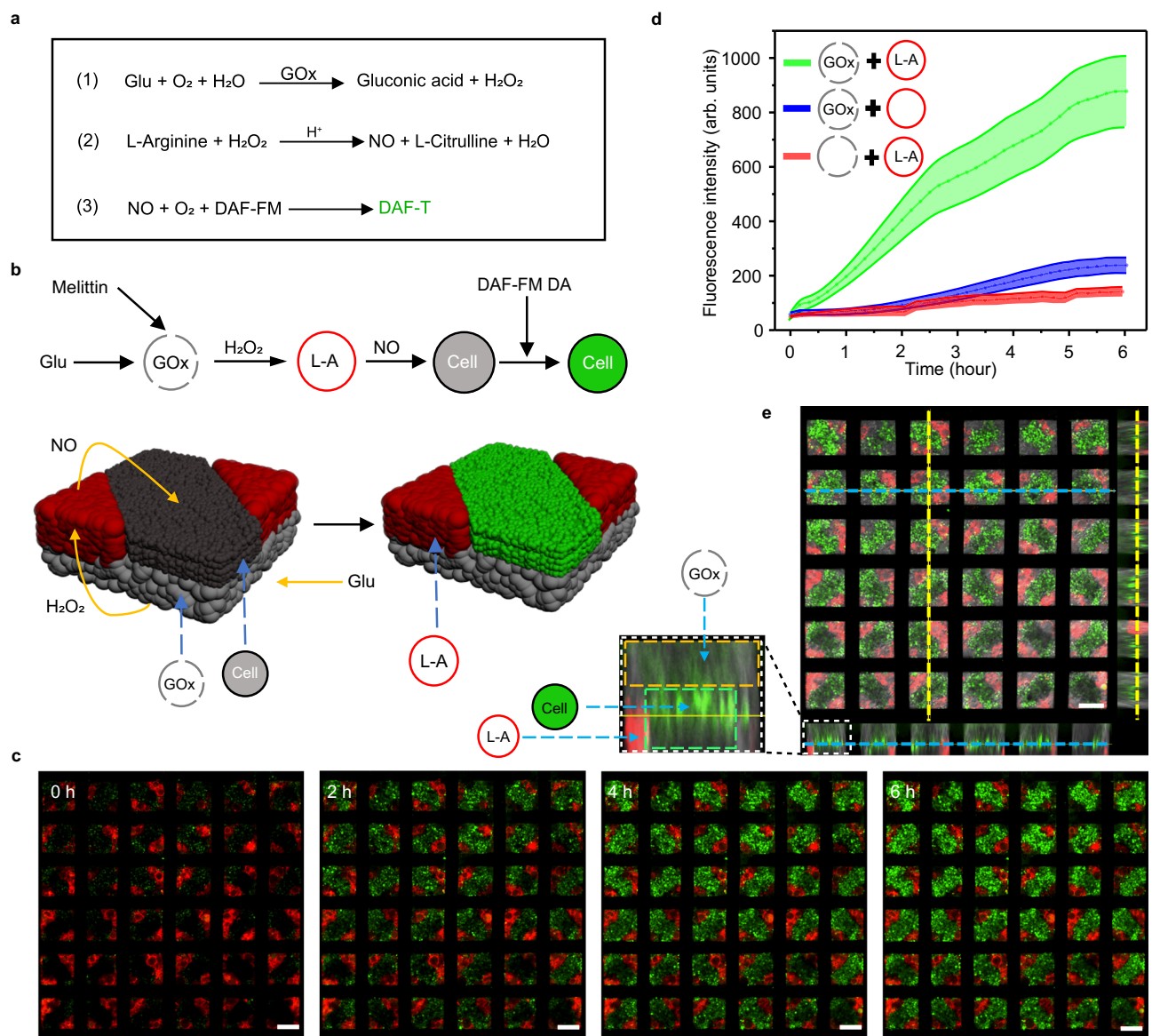

**Fig. 4 Signal communications in hybrid three-component prototissues composed GUVs and C6 glioma cells. a** The cascade reaction formulas for the ternary signal communications among prototissues. Glu, GOx, NO and L-A indicated glucose, glucose oxidase, nitric oxide, and L-Arginine, respectively. **b** Schematic illustration of signal communications among these three components in the prototissues with GOx-GUVs at the bottom, C6 glioma cells at the protruded top layer, and Arginine-rGUVs (containing 20 mM L-Arginine (L-A), pH = 6) at the rest regions of top layer. **c** Fluorescence images (from three independent samples) of the prototissues as a function of time triggered by the addition of glucose (30 mM) in the solution. The red and green regions indicated the Arginine-rGUVs and the cell populations, respectively. **d** The corresponding green fluorescence intensities of prototissues in **c** (green line), same prototissues but using Arginine-free GUVs to replace Arginine-rGUVs (blue line), and the same prototissues but using GUVs with no GOx to replace GOx-GVUs (red line). The error bars were the standard deviation (SD, $n = 5$). **e** A confocal fluorescence image with projected images of the hybrid prototissues composed of three components at 6 h after the addition of glucose. The gray, red, and green regions indicated the non-labeled GOx-GUVs at bottom layer, the Arginine-rGUVs at top edge layer, and the protruded cell at top layer. The scale bars were 100 μm.

treatment (Supplementary Fig. 13). Although the upper permeability limit was not measured, the molecules smaller than 20 kDa were allowed to enter the interior of prototissues (Fig. 5b). The signal communications among the detached prototissues composed of two components (Fig. 5c, d) were investigated to produce NO. It is well known that hydroxyurea (Ha) is oxidized by $H_2O_2$ to generate NO in the presence of HRP[29]. The prototissues composed of non-labeled GOx-GUVs with melittin pores and HRP-rGUVs were fabricated (Fig. 5d, bottom). After the addition of hydroxyurea (10 mM) and glucose (20 mM), a rapid increasing green fluorescence in the prototissues was observed due to the

fluorescence product of DAF-2T generated by the interaction of NO and DAF-2 (10 μM) in the solution (Fig. 5e, pink line in f). As the diffusion of NO from prototissue, the fluorescence intensity in the exterior region was increasing against time (Fig. 5e, the blue line in f), which confirmed the NO-generating capacity of the prepared prototissues. The advantage of prototissues containing GOx-GUVs and HRP-GUVs is that NO can be produced continuously in the presence of glucose and hydroxyurea. While the prototissues containing GOx-GUVs and Arginine-GUVs only produce limited NO due to the amount of arginine inside prototissues.

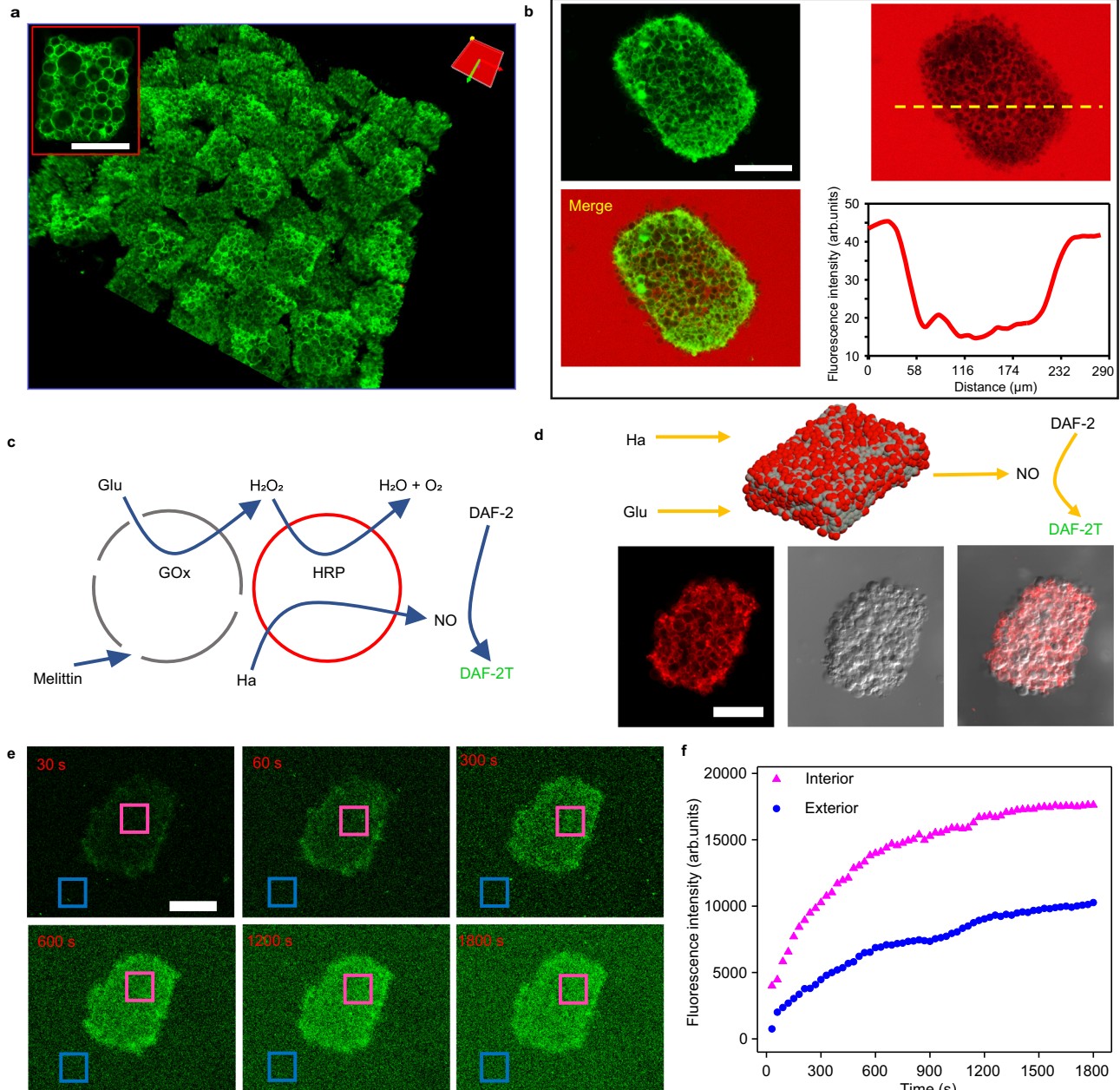

**Fig. 5 Prototissues capable of producing NO. a** The 3D reconstructed confocal image of the GUVs prototissues detached from the NM. The red box indicated an enlarged GUVs prototissue. **b** Fluorescence images of the detached prototissue in Rhodamine-dextran (20 kDa) solution. The yellow dashed line in the fluorescence image correspond to line intensity analysis. **c** Schematic illustration of signal communcations among NO-prototissue composed of GOx-GUVs with melittin pores and HRP-rGUVs. The addition of glucose and hydroxyurea triggered the reaction to generate NO in HRP-rGUVs. NO was detected by the DAF-2 in the solution to generate green fluorescent DAF-2T. Glu, GOx, HRP, Ha and NO indicated glucose, glucose oxidase, horseradish peroxidase, hydroxyurea, and nitric oxide, respectively. **d** Schematic illustration of the structure of the NO-prototissue (top). Fluorescence image of the HRP-rGUVs in the NO-prototissue (bottom left), bright-field image (bottom middle) and their merged image (bottom right) of the NO-prototissue. **e** Time-dependent fluorescence microscopy images of the NO-prototissue in the solution containing DAF-2 (10 μM) after adding glucose (20 mM) and hydroxyurea (10 mM). The green fluorescence channel responded to NO production. **f** Plots of fluorescence intensity against time for internal (pink box) and external regions (blue box) in **e**. The representative fluorescence images were from three independent samples. The scale bars were 100 μm.

**NO-prototissues for vasodilation in vitro.** The prototissues detached from NM capable of producing NO (NO-prototissues) were further investigated with living blood vessel tissue to prove their potential biomedical applications. The thoracic aorta of the rat was isolated from the body and cut into aorta rings with the length of 5 mm, which were subsequently mixed with NO-prototissues in vitro in the organ bath. The schematic illustration of the vascular ring experimental setup was provided in

Supplementary Fig. 14. With the addition of hydroxyurea, NO-prototissues produce NO molecules, which cause blood vessel relaxation (Fig. 6a). A thoracic aorta ring test was used to validate the function of NO-prototissues. A 2 mm-wide thoracic aorta ring was hung on the hooks connected to a force transducer in the HEPES solution containing 20 mM glucose. Upon the addition of hydroxyurea (10 mM) at 15 min, NO-mediated vasodilation was immediately reflected by the force sensor, which

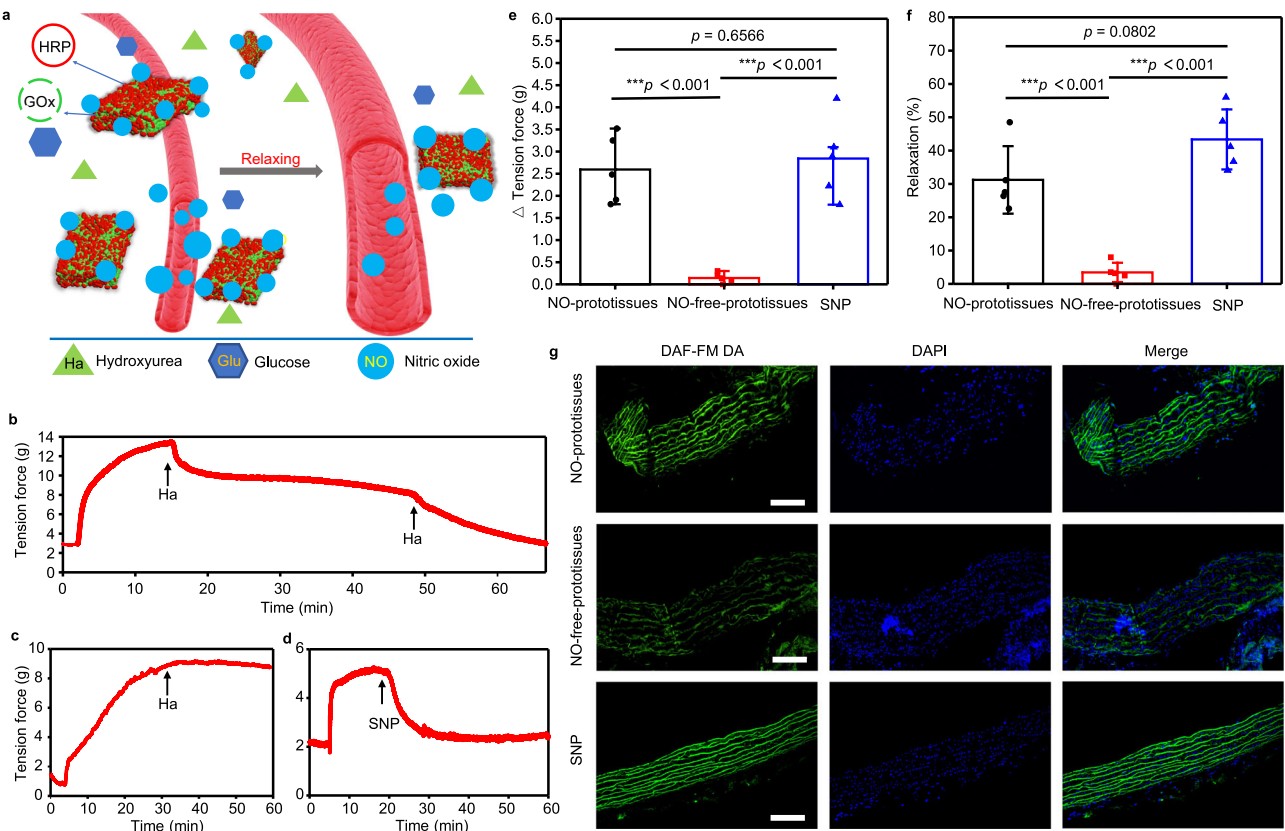

**Fig. 6 NO-prototissues for vasodilation. a** Schematic illustration of vasodilation induced by NO from NO-prototissues. Representative tension curve of vasodilation against time in the presence of NO-prototissues composed with GOx-GUVs and HRP-GUVs (**b**), NO-free-prototissues composed of GOx-free GUVs and HRP-GUVs (**c**) and SNP (**d**). The black arrow indicated the point of the addition of hydroxyurea (Ha, 10 mM) or SNP (0.1 μM). Glu, GOx, HRP, Ha, NO and SNP indicated glucose, glucose oxidase, horseradish peroxidase, hydroxyurea, nitric oxide, and sodium nitroprusside, respectively. **e, f** Bar charts of the decrease in tension force and relaxation of vascular rings observed in the presence of NO-prototissues (2.59 ± 0.77 g; 31.2 ± 10.1% relaxation, black columns), NO-free-prototissues (0.14 ± 0.12 g; 3.4 ± 2.9% relaxation, red columns), and 0.1 μM SNP (2.84 ± 0.92 g; 43.3 ± 9.0% relaxation, blue columns). Data are presented as mean values ± SD, $n = 5$. Statistical analyses were carried out by unpaired two-tailed student's $t$-test (***$p < 0.001$). NO-prototissues: SNP: $p = 0.6566$ in **e** and $p = 0.0802$ in **f**. $p < 0.05$ was considered statistically significant. Source data are provided as a Source Data file. **g** Fluorescence images (from three independent samples) of vascular sections stained with DAF-FM DA (the green fluorescence channel responded to NO production, left column), DAPI (blue fluorescence with nuclei, middle column), and their merge images (right column). The blood vessels were treated using NO-prototissues (top row) or NO-free-prototissues (middle row) in the HEPES solution containing 10 mM hydroxyurea for 20 min, respectively. The blood vessel was treated using 0.1 μM SNP in the HEPES solution for 20 min (bottom row). The scale bars were 100 μm.

exhibited a rapid decrease of tension force (Fig. 6b). For comparison, there was no obvious change in the tension force when hydroxyurea (10 mM) was added to the solution containing NO-free-prototissues composed of GOx-free GUVs and HRP-GUVs at 30 min (Fig. 6c) because NO-free-prototissues did not produce NO molecules. On the contrary, sodium nitroprusside (SNP) as the common NO donor was used to treat aorta ring[30,31]. A rapid decrease of tension force was observed immediately upon the addition of 0.1 μM SNP at 20 min (Fig. 6d). The variations of the tension force were calculated before and 10 min after the addition of hydroxyurea. Typically, NO-prototissues resulted in a tension force decrease of 2.59 g and a blood vessel relaxation of 31.2% (Fig. 6e, f, black columns). When the blood vessels were treated with the NO-free-prototissues, there was a small tension force decrease of 0.14 g and a blood vessel relaxation of 3.4%, which was attributed to the low-level degradation of hydroxyurea in the media (Fig. 6e, f, red columns). With the treatment of 0.1 μM SNP, the tension force decreased by 2.84 g with the blood vessel relaxation of 43.3% (Fig. 6e, f, blue columns). Moreover, the sequential addition of the same amount of hydroxyurea at 48 min (Fig. 6b) induced the blood vessels to relax again, which

confirmed that the blood vessels remained viable within the experimental period. Fluorescence images of blood vessels sections also confirmed the formation of NO in the vascular ring through a cell-permeable NO dye (DAF-FM DA) (Fig. 6g). NO-prototissues and 0.1 μM SNP induced high-intensity NO-mediated green fluorescence in blood vessels (Fig. 6g, top row and bottom row). On the contrary, lower fluorescence intensity was observed in NO-free-prototissues mediated experiments due to the small amount of degraded hydroxyurea (Fig. 6g, middle row). These experimental results implied the potential application of NO-prototissue in cardiovascular disease.

In summary, the phospholipid vesicles and C6 glioma cells were used as the building blocks to precisely fabricate high-throughput spatial coded multicomponent prototissues using the magneto-Archimedes effect. The signal communications were demonstrated in the spatially programmable two- and three-component prototissues on the NM using two-step and three-step enzyme cascade reactions, respectively. Significantly, the prototissues detached from the NM composed of GOx-GUVs and HRP-GUVs produced NO via the internal communications in the prototissues triggered by glucose and hydroxyurea. These

prototissues were confirmed to be capable of relaxing the rat blood vessels, consequently improving their functions, which may hold the potential to treat cardiovascular diseases.

## Methods

**Materials**. 1,2-Dioleoyl-sn-glycero-3-phosphocholine (DOPC), Texas Red-labeled 1,2-dihexadecanoyl-sn-glycero-3-phosphoethanolamine triethylammonium salt (TR-DHPE), and N-(7-nitrobenz-2-oxa-1,3-diazol-4-yl)-1,2-dihexadecanoyl-sn-glycero-3-phosphoethanolamine triethylammonium salt (NBD-PE) were purchased from Avanti Polar Lipids (USA). Horseradish peroxidase (HRP), glucose oxidase (GOx), amplex red, melittin, hydroxyurea (Ha), 4-aminomethyl-2′,7′-difluorofluorescein diacetate (DAF-FM DA), 2-(3,6-dihydroxy-4,5-diamino-9H-xanthen-9-yl)-benzoic acid (DAF-2), DAPI and L-Arginine (L-A) were purchased from Sigma (USA). Sucrose (Suc), glucose (Glu), Manganese (II) (MnCl$_2$) chloride, Calcium chloride (CaCl$_2$), and hydrochloric acid (HCl) were obtained from Aladdin (China). Gadobutrol injections were obtained from Harbin Institute of Technology Hospital (China). The nickel meshes (NM) were purchased from Gates RGRS Company (China). Cylindrical NdFeb magnets (1 T, diameter = 3 cm, thickness = 1 cm) were bought from Gates Qiangci Company (China). The Indium tin oxide (ITO) electrodes were obtained from Hangzhou Yuhong technology Co. Ltd (China). Dulbecco's modified Eagle's medium (DMEM), phosphate buffer saline (PBS) without calcium and magnesium, trypsin, streptomycin, and penicillin were purchased from Corning (USA). The fatal bovine serum (FBS) was purchased from Gibco (USA).

**Preparation of GUVs**. The giant unilamellar vesicles (GUVs) were prepared by electroformation method (Supplementary Fig. 15)[32–34]. In brief, DOPC (5 mg/mL, 8 µL) mixed with the fluorescent lipids TR-DHPE (0.5% w/w, red fluorescence) or NBD-PE (5% w/w, green fluorescence) in chloroform was gently spread on two ITO electrodes to form lipid films. A rectangular polytetrafluoroethylene frame was placed between two lipid film-coated ITO electrodes to form the setup. An AC electric field (5 V, 10 Hz) was applied by a signal generator for 2 h, followed by applying an electric field (0.8 V, 2 Hz) for an additional 5 min. Different solutions were filled inside the frame to prepare the wanted GUVs. The sucrose solution (300 mM) containing 20 µg/mL HRP or 30 µg/mL GOx was used to obtain HRP-GUVs or GOx-GUVs. The sucrose solution (280 mM) containing 20 mM L-Arginine (pH = 6) was used to prepare Arginine-GUVs. The free GOx, HRP, and L-Arginine in the supernatants were removed by centrifugation (410 × g, 5 min, 5 times) until no corresponding molecules were detected in the supernatants (Supplementary Fig. 16). The GUVs were incubated in a solution containing 50 µg/mL melittin for 20 min to obtain the GUVs with melittin pores in the lipid bilayer membrane. The initial concentration of GUVs ((2.50 ± 0.18) ×10$^8$/mL) was obtained using flow cytometry (Supplementary Fig. 17).

**Formation of the prototissues**. Prototissues were prepared using a homemade device (Supplementary Fig. 1). A square NM (1.2 × 1.2 cm) was fixed in a petri-dish using vacuum grease. To avoid GUVs rupture during assembly, 200 µL of 0.1 mg/ml DOPC ethanol-water solution with an ethanol volume percentage of 40% was added to the petri-dish at 45 °C for 10 min, followed by washing the device with MnCl$_2$ solution or PBS for three times. For the formation of prototissues under a vertical magnetic field, the device was placed on the center of the top surface of a circular permanent magnet, followed by adding GUVs labeled with NBD-PE (gGUVs) or TR-DHPE (rGUVs) in 100 mM MnCl$_2$ solution. The inclined magnetic field was provided by putting the NM at the edge regions of the top surface of the magnet as shown in Fig. 1f. For the formation of two-component prototissues (Fig. 2b–d, f and g), gGUVs and rGUVs were successively added to the device under the vertical or inclined magnetic field. The two-component prototissues in Fig. 2e were obtained by adding the mixture solution of gGUVs and rGUVs into the device under the vertical magnetic field. For the formation of three-component prototissues in Fig. 2h, non-labeled GUVs and gGUVs were successively trapped under the vertical magnetic field, and then rGUVs were trapped in the absence of the magnetic field. After one type of GUVs was trapped, the time intervals were 1 h before adding another type of GUVs. The concentration of GUVs was controlled by varying the volume ratio of the GUVs sucrose solution and MnCl$_2$ (gadobutrol) solution. Before preparing prototissue array containing living cells, the device and the solution were sterilized and the MnCl$_2$ solution was replaced by PBS solution containing 30 mM gadobutrol. The device was placed at 37 °C and 5% CO$_2$ in a humidified atmosphere throughout the experiments containing cells.

**Cell line and cell culture**. C6 glioma cells were cultured in DMEM containing 50 µg/mL penicillin, 50 µg/mL streptomycin, and 10% fetal bovine serum at 37 °C and 5% CO$_2$ atmosphere. The cells were digested by trypsin, concentrated by centrifugation at 410 × g for 5 min, and redispersed in the medium with the final density of 2 × 10$^5$/mL.

**Signal communications in prototissues**. The GOx-gGUVs with melittin pores and HRP-GUVs were magnetically assembled into the structure as shown in Fig. 3b. To initiate the signal communication, the external solution was replaced with 30 mM glucose containing 0.05 µM Amplex Red. The fluorescent product was

monitored using a fluorescence microscope. For comparison, GOx-free gGUVs were used to replace the GOx-gGUVs in the prototissues. To form the prototissue as shown in Fig. 4b, the GOx-GUVs with melittin pores and C6 glioma cells were successively trapped under the vertical magnetic field, and then Arginine-rGUVs (containing 20 mM L-Arginine) were trapped without magnetic field in the PBS containing 30 mM gadobutrol. C6 glioma cells were incubated in PBS containing NO probe (10 µM DAF-FM DA) for 20 min before adding them to the device. To initiate the signal communication, the external solution was replaced with PBS containing 30 mM glucose. For comparisons, GOx-GUVs or Arginine-rGUVs in the prototissue were replaced by GOx-free GUVs or Arginine-free rGUVs, respectively. The prototissues were on the NM during the signal communications.

**Detachment of prototissues capable of producing NO**. After the GUVs prototissue array was assembled on the NM, 100 mM CaCl$_2$ was used to promote hemi-fusion of the prototissues composed of non-labeled GOx-GUVs with melittin pores and HRP-rGUVs for 10 min. The prototissues were detached from the NM by gently shaking in the solution. We named these prototissues as NO-prototissues. Afterward, the prototissues were washed three times using PBS to remove CaCl$_2$ in the solution. The square nickel meshes containing 2116 grids were used to prepare NO-prototissues. Each nickel meshes produced about 1938 NO-prototissues. The permeability of prototissue detached from the NM was detected using Rhodamine-dextran (20 kDa). Ten millimolar hydroxyurea and 20 mM glucose were added to the device to trigger the reaction. Ten micromolar DAF-2 was used to detect the released NO from prototissues detached from the NM.

**Vasodilation stimulated by NO from NO-prototissues in vitro**. The thoracic aortas of rats were separated after the rats were anesthetized (n = 5 vascular rings per group, 15 vascular rings, 8 rats). The blood vessels were cut into rings after the extraneous fat and connective tissues were removed mechanically. The rings were connected to a force transducer in the baths containing 10 mL of HEPES solution (component (mM): NaCl 137.6, KCl 4.0, MgCl$_2$ 1.1, CaCl$_2$ 2.1, NaHCO$_3$ 10.0, NaH$_2$PO$_4$ 0.92 and Glucose 20, pH 7.4) at 37 °C under physiological O$_2$ conditions. After equilibration for 20 min, a solution with high potassium concentration (KCl, 70 mM) was used to induce the contraction of the vascular rings. When the tension force reached equilibrium, about 1 × 10$^4$ NO-prototissues detached from the 5 NMs composed of GOx-GUVs with melittin pores and HRP-GUVs and 10 mM hydroxyurea were added into the bath solution. The tension force was detected by the force transducer. For comparison, the NO-free-prototissues detached from NM composed of GOx-free GUVs and HRP-GUVs were used to replace NO-prototissues. The amount of variation in tension force (ΔTension) and the percentage of relaxation (Relaxation %) were defined as follows: ΔTension force = Fa − Fb; Relaxation % = (Fa − Fb)/ Fb × 100%. Fb and Fa represented the tension force before and after adding the prototissues and hydroxyurea, respectively. Before the vascular rings were snap-frozen and cut into slices, they were incubated for 20 min in the PBS containing 10 µM DAF-FM DA after being treated with the prototissues. We confirmed that ethical approval from the Experimental Animal Ethics Committee of Harbin Institute of Technology (IACUC-2021012) was obtained prior to the study.

**Instruments**. The prototissues were characterized by an inverted fluorescence microscope (Olympus IX73, Japan), laser confocal microscope (Olympus FV 3000, Japan), and upright fluorescence microscope (Nikon 80i, Japan). The AC electric field was generated with a signal generator (Agilent 33220a-001, USA). The magnetic field was simulated using COMSOL Multiphysics 5.4 software. The concentration of GUVs was estimated using flow cytometry (BD FACSAriaIII flow cytometer, USA). The tension force was detected by a tension sensor in a constant temperature perfusion system for isolated tissues and organs (TECHMAN HV1403, China). Fluorescence microscope images were analyzed by CellSens 2.1 and ImageJ 1.8.0. The data process was conducted using Origin 9.0 and Office 2019.

**Reporting summary**. Further information on research design is available in the Nature Research Reporting Summary linked to this article.

## Data availability
Source data are provided with this paper.

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

## Acknowledgements

This work was supported by the National Natural Science Foundation of China (Grant No. 21929401 and 21773050 to X.J.H.) and the Heilongjiang Touyan Team (HITTY-20190034 to X.J.H.).

## Author contributions

X.J.H. supervised the research. X.J.H., X.X.Z., and C.L. conceived and designed the experiments. X.X.Z., C.L., F.K.L., Y.S.R., B.Y.Y., and W.M. performed experiments. X.J.H., X.X.Z., C.L., F.K.L., Y.S.R., B.Y.Y., and W.M. analyzed the data. X.J.H., X.X.Z., and C.L. wrote the paper. All authors commented on the paper. X.X.Z. and C.L. contributed equally to this work.

## Competing interests

The authors declare no competing interests.
