## [Peer Review File · Nature Communications]

High-throughput production of functional prototissues capable of producing NO for vasodilationREVIEWER COMMENTS

Reviewer #1 (Remarks to the Author):

In this manuscript the authors further their recent efforts in fabricating vesicle-based prototissues using the magneto-archimedes effect. Here, they use this guided assembly method to build 3D prototissues containing multiple types of GUV, as well as hybrid tissue architectures composed of GUVs and glioma cells.

By encapsulating different content within different sections of the prototissue, the authors implement multistep (bio)chemical reactions similar to those implemented in their recent Nature Communications publication. A key difference between this manuscript and previous work is connecting novel and relevant external inputs (e.g. hydroxyurea) to generate new reaction sequences in the prototissue - ultimately producing nitric oxide (NO) - and connecting this to a biological system in order to control its behaviour. NO has potential biomedical applications from vasodilation to cancer, and NO production here is coupled to the former, with NO-producing prototissues successfully causing the relaxation of a mouse aortic ring. To facilitate these final experiments, the authors also demonstrate the removal of prototissues from their production grids, a critical step in characterising free-standing GUV-prototissues as well as facilitating translation of tissues to various applications.

I enjoyed reading this manuscript and the work is both of high quality and novelty, elegantly highlighting how reaction sequences can be implemented in prototissues and non-living structures can be interfaced with, and influence, living cells in the same prototissue (glioma hybrid tissue) or with natural tissue (aortic rings). Such work will be of great interest to researchers in many fields including bottom-up synthetic biology, bioengineering and regenerative medicine. The manuscript is generally well written, and the claims of the authors are supported by the experiments highlighted in the manuscript and the supplementary information. This work possesses the quality, impact and multidisciplinary-focus necessary to be published in Nature Communications once the following comments below have been addressed.

Specific comments:

- How were the GUV concentrations estimated? I could not see any description of quantification method in manuscript.
- Line 103/4 - GUV concentrations used for fabricating prototissues listed but no panels listed - e.g. " 3×10^5 /ml for , 5×10^5 /ml for ," please clarify how this information is presented in figure 1C in the manuscript and in the figure caption.
- Line 161 - for three component prototissues where the third component is added without the presence of the magnetic field, how does the stability of this third section compare to the first two assembled in the magnetic field? Without the presence of the field are the same close packing arrangements of GUVs present in the tissue? It would be interesting to see how these three-component tissues behave under different osmotic stress conditions - integrity of each tissue section is critical if similar production strategies are to be pursued in the future.
- Fig3A. Is amplex red added to external solution similarly to glucose? This should be made clearer in the manuscript text if so. Why was amplex red added externally and not encapsulated in the HRP GUVs?
- Fig 3E. - It looks like the control experiment (-gOX) has no error bars? Are they present but small? It is hard to tell in the figure. If they are not present can you please include.
- Line 222 - Have you performed any cell viability experiments for three component hybrid prototissues? It would be interesting to know the viability over time. Do glioma cells grow in the prototissue structure?

- Line 237 – typography error – GOx-GVUs instead of GOx-GUVs
- Line 259 – What happens when the prototissues are detached without CaCl₂ washing? How long do the prototissues survive once detached from their production grids?
- Line 261 – How do you know that molecules larger than 20kDa could not penetrate the prototissue? Did you test the tissue permeability of larger fluorescent dextrans? If you did not please amend the manuscript to state that the upper permeability limit has not been measured.
- Fig 5 – why did you use a hydroxyurea + HRP approach to generating NO here instead of using the arginine system in figure 4? It would be good to discuss this change of strategy in the manuscript – what is the advantage of encapsulating HRP over arginine in prototissues?
- Line 293 – HEPES? Should be HEPES?
- For vasodilation experiments, the concentration of prototissues is listed as 1x10⁴? How did you calculate this tissue concentration?
- In fig 6, how close were the prototissues to the aorta? Could you provide images of the experimental setup, or additional schematics that show the distance between aorta and prototissue? It is hard to visualise from the cartoon schematic.
- Fig 6 – spelling errors – relaxating, hadroxyurea
- In your conclusion you briefly mention the use of prototissues in transplantation applications. Could you briefly expand on how prototissues could be used in this or other biomedical applications? Are there other potential applications for the prototissues generated here?

Reviewer #2 (Remarks to the Author):

The work of Prof Han and coworkers describe a spatially organized system of prototissues, assembled from two or more type of giant unilamellar vesicles GUV vesicles. The assembly is succeeded in a paramagnetic solution and with the assistance of an external magnet and a nickel mesh placed underneath the assembly of the mesh. They also nicely demonstrate two-step and three-step signal communications in the prototissues using cascade enzyme reactions. In such a demonstration they produce NO and use this for biological response in a blood vessel tissue. The paper is well organized and clearly written. The claims are supported by the data provided. The group has already shown in a previous paper published in Nat Comm ref. 13 (<https://doi.org/10.1038/s41467-019-14141-x>) that they can nicely assemble in 3D the GUVs. Based on that I see the advance in this paper is mostly on the demonstrations where they used the assemblies for the cascade reactions and so on. These demonstrations are good and convincing, however, I donot see why and how the prototissues achieved here will be beneficial for the advance of the field. What I mean here; is do these prototissues have to be well programmed and submillimeter sized to be able to function as in these demos. Because authors perform their experiments in Fig 5 and Fig 6 in solutions and these experiments will indeed give similar results even if the authors would use a pellet of bulk assemblies of the same GUVs, e.g. that is obtained via sedimentation of the same types of GUVs sequentially to obtain an assembly of the GUVs on top of eachother. The possibility of having sub-milimeter programmed assemblies of GUVs does not seem to me the only way to obtain the reactions of the blood vessel or the cascade reactions shown in the paper. These reasons make me wonder if this follow up paper has built enough advance on the previous work to qualify being published at Nature Comm.

There is a typo in line 299, it is written 'molecure' instead of molecule.

Reviewer #3 (Remarks to the Author):

This manuscript presents an interesting novel technology, built upon recent advancements in engineering synthetic cells and tissues. The writing is mostly clear, results are presented and analyzed in convincing and conclusive way.

Introduction:

In the first paragraph of the results and discussion section, the magneto-Archimedes effect and the composition of the prototissues are explained, however it would be helpful to mention this early in the introduction as these are key to understanding the background of the paper. The rest of the introduction is successful in orienting the reader to what will be described in the paper.

Results and Discussion:

A minor comment, but when describing the magnetic field regions on the nickel mesh (NM) structures, it is mentioned that the weak magnetic field regions are the blue areas in Fig 1b – there are multiple blue areas and it would be helpful to specify that the weak regions are the light and/or the dark blue regions. However, the rest of the figures are useful in visualizing the results and have clear explanations.

In the section “Diverse multi-compartment prototissues”, it is mentioned that assembling the multi-component prototissues can be attained by adjusting the distribution of the magnetic fields. Including a reference to the top box of Figure 2 here would help explain the differences in these fields. As described in this section, the level of control in the formation and structure of the prototissues is impressive.

I wonder if an interesting addition to the multi-component prototissue communication assays would be including an experiment where the communication GUVs (i.e. Gox-GUVs) are free-floating in buffer to understand how they may communicate less/more effectively when in solution than when in the prototissue structure.

In the section “Prototissues capable of producing NO,” was there a test to see if/how many GUVs are dissociating from the proto-tissue structure? Measuring the fluorescence of the buffer in the petri-dish over time would be a good way to measure if GUVs encapsulating these different components are dissociating from the proto-tissues.

It is very exciting to see that the prototissues can relax blood vessels by release of NO in the section “NO-prototissues for vasodilation.” It should be clarified here that this work was done in vitro. The authors write that the NO-prototissues are “mixed” with living blood vessels, can they elaborate? What is the spatial relationship of the prototissues with the blood vessels—are they free floating in solution together?

In Figure 6, it is written that the blood vessels were treated using NO-prototissues and NO-free-prototissues. Including images of the blood vessels treated by pure NO in solution (not GUVs) would be more rigorous.

Finally, in the summary, the authors write that these prototissues could be used to treat cardiovascular diseases through transplantations. I think this is a thrilling and plausible speculation based on the work described in this paper. Can the authors speculate on how the prototissues may be bound to distinct areas in the body, and how long would these GUVs be able to relax the vessels?

Methods:

Are the prototissues on and off the NM stored in buffer in the petri-dish? I think this would be important to include in the results section, as well as methods. It should again be clarified in the methods that vasodilation stimulated by NO from NO-tissues was done in vitro.

Response to the reviewers' comments

For the sake of clarity, the comments of the reviewer have been collated in black and *italic*, and our response to each comment appears in blue. All the changes to the manuscript are highlighted in red.

Reviewer #1 (Remarks to the Author):

In this manuscript the authors further their recent efforts in fabricating vesicle-based prototissues using the magneto-archimedes effect. Here, they use this guided assembly method to build 3D prototissues containing multiple types of GUV, as well as hybrid tissue architectures composed of GUVs and glioma cells.

By encapsulating different content within different sections of the prototissue, the authors implement multistep (bio)chemical reactions similar to those implemented in their recent Nature Communications publication. A key difference between this manuscript and previous work is connecting novel and relevant external inputs (e.g. hydroxyurea) to generate new reaction sequences in the prototissue - ultimately producing nitric oxide (NO) – and connecting this to a biological system in order to control its behaviour. NO has potential biomedical applications from vasodilation to cancer, and NO production here is coupled to the former, with NO-producing prototissues successfully causing the relaxation of a mouse aortic ring. To facilitate these final experiments, the authors also demonstrate the removal of prototissues from their production grids, a critical step in characterising free-standing GUV-prototissues as well as facilitating translation of tissues to various applications.

I enjoyed reading this manuscript and the work is both of high quality and novelty, elegantly highlighting how reaction sequences can be implemented in prototissues and non-living structures can be interfaced with, and influence, living cells in the same prototissue (glioma hybrid tissue) or with natural tissue (aortic rings). Such work will be of great interest to researchers in many fields including bottom-up synthetic biology, bioengineering and regenerative medicine. The manuscript is generally well written, and the claims of the authors are supported by the experiments highlighted in the manuscript and the supplementary information. This work possesses the quality, impact and multidisciplinary-focus necessary to be published in Nature Communications once the following comments below have been addressed.

Specific comments:

1. How were the GUV concentrations estimated? I could not see any description of quantification method in manuscript.

Thank the reviewer for the comments. The initial concentration of GUVs ($(2.50 \pm 0.18) \times 10^8$ /mL) was obtained using flow cytometry. The other desired GUV concentration

was estimated by diluting the initial GUV solution.

We added below sentences in page 16 and 18 of the manuscript.

Page 16, ‘The initial concentration of GUVs ($(2.50 \pm 0.18) \times 10^8$ /mL) was obtained using flow cytometry.’

Page 18, ‘The concentration of GUVs was estimated using flow cytometry (BD FACSAriaIII flow cytometer, USA).’

2. Line 103/4 – GUV concentrations used for fabricating prototissues listed but no panels listed – e.g. “ 3×10^5 /ml for , 5×10^5 /ml for ,” please clarify how this information is presented in figure 1C in the manuscript and in the figure caption.

Thank the reviewer for the comments. We have corrected the miscodes in the manuscript and the figure caption in page 3-6 as below.

Page 3, ‘As expected, the GUVs were firstly assembled in the weak magnetic field regions (dark blue areas in Fig. 1b, bottom layer) inside each grid (Fig. 1c I). With the number of added GUVs increasing, GUVs gradually filled the space inside each grid of the NM to generate 60 μ m thick bottom layer (Fig. 1c II) and further protruded to form 140 μ m thick brick-shape top layer above the GUVs aggregations inside each grid (Fig. 1c III, IV), which were consisted with simulated field distribution (dark blue areas in Fig. 1b, top layer). The concentrations of the GUVs for fabricating the prototissues in Fig. 1c were 3×10^5 /mL for I, 5×10^5 /mL for II, 8×10^5 /mL for III and 1.2×10^6 /mL for IV, respectively.’

Page 5, ‘c Schematic and fluorescence images of the prototissues assembled in a vertical magnetic field. With the number of added green giant unilamellar vesicles (gGUVs) increasing, the prototissues changed from single layer (I, II) to double layers (III, IV).’

Page 6, ‘With a vertical field, the structures similar to Figure 1d IV but composed of rGUVs (6×10^5 /mL) and gGUVs (6×10^5 /mL) were obtained with the addition of the mixture of rGUVs and gGUVs (Fig. 2e).’

3. Line 161 – for three component prototissues where the third component is added without the presence of the magnetic field, how does the stability of this third section compare to the first two assembled in the magnetic field? Without the presence of the field are the same close packing arrangements of GUVs present in the tissue? It would be interesting to see how these three-component tissues behave under different osmotic stress conditions – integrity of each tissue section is critical if similar production strategies are to be pursued in the future.

Thank the reviewer for the comments. The third section was stable even if the GUVs were added without the presence of magnetic field according to below two experimental

results. 1) We did not observe the detachment of GUVs after the nickel mesh was inverted for 120 mins (Supplementary Fig.7a). 2) The three-component prototissues can be kept intact at least 18 days (Supplementary Fig.7b). The density of the GUVs assembled without the presence of the magnetic field is $6353 \pm 1066 / \text{mm}^2$, which is lower than that in the presence of the field ($10571 \pm 1378 / \text{mm}^2$) (Supplementary Fig. 8g). The magnetic field enabled the dense packing of GUVs. The three-component tissues behavior under different osmotic stress conditions were investigated in details (Supplementary Fig. 8). The experimental results and corresponding descriptions were added into the manuscript as below.

We added below contents in page 6 of the manuscript.

Page 6, ‘The three-component GUVs prototissues were stable on the nickel mesh when it was inverted for 120 mins (Supplementary Fig. 7a). The three-component prototissues were maintained intact for 18 days (Supplementary Fig. 7b).’.

Supplementary Fig. 7 Stability of the three-component prototissues. (a) Fluorescence images of the prototissues on the nickel mesh being inverted for 1 min (top row) and 120 mins (bottom row) respectively. Green GUVs populations assembled with magnetic field (left column), red GUVs populations assembled without magnetic field (middle column) and their merged images (right column). (b) Fluorescence images of prototissues on the nickel mesh at different days. The scale bars were $100 \mu\text{m}$.

We added below sentences in page 8 of the manuscript.

Page 8, ‘Behaviors of the three-component prototissues under different osmotic

stress conditions. For the prototissues composed of GUVs encapsulating 300 mM sucrose, a hypotonic condition ($\Delta\Pi = 743.3$ kPa) was created by replacing the solution with pure water. The areas in yellow boxes (at hypotonic condition, Supplementary Fig. 8a2) became larger than those in cyan boxes (at isotonic condition, Supplementary Fig. 8a1), which indicated the expansion of green GUVs populations (assembled with magnetic field). Meanwhile the tissue volume increased (Supplementary Fig. 8a4 and a5), because the height of the projected image at hypotonic condition (yellow box in Supplementary Fig. 8a4) was larger than that at isotonic condition (cyan box in Supplementary Fig. 8a5) at the same cross sections. By overlapping the cyan boxes and yellow boxes (Supplementary Fig. 8a3), the expansion percentage of GUVs populations was 16.41% (Supplementary Fig. 8d, red box) from $(A_a - A_b)/A_b \times 100\%$, where A_a and A_b were the average area under hypotonic and isotonic conditions respectively. Similarly, the prototissue volume increased by 35.28% according to the heights of cyan and yellow boxes in Supplementary Fig. 8a6. The areas of red GUVs populations decreased by 26.99% (Supplementary Fig. 8e, red box) due to the confinement of the nickel mesh grids. At isotonic condition, the density of the green GUVs assembled with magnetic field was $10571 \pm 1378/\text{mm}^2$, which was 1.66 times that of red GUVs assembled without magnetic field ($6353 \pm 1066/\text{mm}^2$) (Supplementary Fig. 8g). At hypotonic condition, the expanded green GUVs populations ($7628 \pm 673/\text{mm}^2$) pushed the red GUVs into close packed structures ($6904 \pm 1022/\text{mm}^2$) (Supplementary Fig. 8g). On the contrary, the area of the green GUVs populations shrank by 7.41% (Supplementary Fig. 8b1, b2, b3 and 8d green box) and 18.40% (Supplementary Fig. 8c1, c2, c3 and 8d purple box) under hypertonic conditions when the solution was replaced with 600 mM ($\Delta\Pi = -743.3$ kPa) and 900 mM ($\Delta\Pi = -1486.6$ kPa) glucose solution, respectively. The red GUVs populations with relative loose structure filled the space caused by the shrinking green GUVs populations, which resulted in the red GUVs populations areas to increase by 13.40% and 61.57%, respectively (Supplementary Fig. 8e, green and purple boxes). The prototissues volume decreased by 26.86% and 31.96% under the hypertonic conditions of 600 mM and 900 mM glucose solution, respectively (Supplementary Fig. 8f, green and purple boxes). The three-component prototissues were stable under hypertonic and hypotonic conditions. They exhibited collective expansion and contraction behaviors.’.

Supplementary Fig. 8 Behaviors of three-component prototissues under different osmotic stress conditions. Behaviors of prototissues under hypotonic condition ($\Delta\Pi = 743.3$ kPa) (**a**) with pure water solution, hypertonic conditions with 600 mM ($\Delta\Pi = -743.3$ kPa) (**b**) and 900 mM ($\Delta\Pi = -1486.6$ kPa) (**c**) glucose solutions. Fluorescence images of the prototissues under isotonic (**a1**, **a4**, **b1**, **b4**, **c1** and **c4**) and under osmotic stress conditions for 30 minutes (**a2**, **a5**, **b2**, **b5**, **c2** and **c5**). The cyan and yellow boxes in **a3**, **b3** and **c3** indicated the areas of the green GUVs populations in the white dashed boxes in **a1**, **b1**, **c1** and **a2**, **b2**, **c2**, respectively. The cyan and yellow boxes in **a4**, **a5**, **b4**, **b6** and **c4**, **c5** indicated projected images of prototissues. The cyan and yellow boxes in **a6**, **b6**, **c6** indicated the tissue height at isotonic and osmotic stress conditions, respectively. Box plots of the percentage of area variation of the green (**d**) and red (**e**) GUVs populations under different osmotic stresses. $\Delta A / A_b = (A_a - A_b) / A_b \times 100\%$, where A_b and A_a represented the areas of the green (or red GUVs) populations at

isotonic and osmotic conditions, respectively ($n = 9$). (f) Box plots of the percentage of volume variations of the prototissues under different osmotic stress. $\Delta V/V_b = (V_a - V_b)/V_b \times 100\%$, where V_b and V_a represented the volume of the prototissues at isotonic and osmotic conditions, respectively ($n = 12$). (g) Box plots of density of the green and red GUVs ($n = 9$). g_b , g_a represented the density of green GUVs at isotonic and hypotonic conditions ($\Delta\Pi = 743.3$ kPa), respectively. r_b , r_a represented the red GUVs at isotonic and hypotonic condition ($\Delta\Pi = 743.3$ kPa), respectively. Vertical center line represented the median. Top and bottom bounds of the box plots represented the first and third quartile. The tips of the whiskers represented min and max values. The scale bars were $100 \mu\text{m}$.

4. Fig3A. Is amplex red added to external solution similarly to glucose? This should be made clearer in the manuscript text if so. Why was amplex red added externally and not encapsulated in the HRP GUVs?

Thank the reviewer for the comments. Amplex red and glucose molecules were added to the external solution. We added the below sentence in page 9 of the manuscript.

Page 9, 'To initiate the reactions, glucose molecules (30 mM) and Amplex Red (0.05 μM) were added into the bath solution simultaneously.'

The reason for the addition of Amplex red molecules externally is because that Amplex red molecules can penetrate lipid bilayer by free diffusion (*PNAS*, **109**, E1437-E1443, 2012). If we encapsulated Amplex red molecules inside HRP-GUVs, they would diffuse out during the period of prototissue preparation.

5. Fig 3E. – It looks like the control experiment (-GOx) has no error bars? Are they present but small? It is hard to tell in the figure. If they are not present can you please include.

Thank the reviewer for the comments. The control experiment has error bars, but they are rather small. They can be seen in the zoom-in figure as shown below.

6. Line 222 – Have you performed any cell viability experiments for three component hybrid prototissues? It would be interesting to know the viability over time. Do glioma

cells grow in the prototissue structure?

Thank the reviewer for the comments. We monitored the cell viability over time in the three-component hybrid prototissues (Supplementary Fig. 11). The fluorescence images in Supplementary Fig. 11a showed the dead glioma cells labeled by propidium iodide (PI) (red fluorescence) over time. The death rate curve of the glioma cells in the tissue as a function of time exhibited similar trajectory to that of the free cells in the PBS (Supplementary Fig. 11b). Glioma cells were almost dead after 24 hours (Supplementary Fig. 11c). Green and red fluorescence indicated the GUVs and dead cell populations, respectively. The projected images of prototissues at 24 hours confirmed the three-component tissues composed of non-labeled GUVs populations (the gray areas) at the bottom layer, gGUVs populations (the green areas) at the edge of top layer, and cell populations (the red areas) in the middle of the top layer.

From above observations, we did not observe cell proliferation instead of death in PBS solution.

We added the below sentence in page 11 of the manuscript.

Page 11, 'The glioma cells in the prototissues showed similar death rate curve to the free cells as a function of time in the PBS solution, which confirmed that the GUVs prototissue did not affect cell viability (Supplementary Fig. 11).'

Supplementary Fig. 11 Viability of the glioma cells in the hybrid prototissues over time. (a) Fluorescence images of the hybrid prototissues over time, where red fluorescence and green fluorescence dots indicated the dead cells labelled by propidium

iodide (5 μM) and green GUVs populations in PBS solution, respectively. **(b)** The death rates of the glioma cells in the hybrid prototissues and the free glioma cells in PBS solution as a function of time, $n = 3$. **(c)** A 3D fluorescence image of the dead cells (red fluorescence) and green GUVs populations in the prototissues in PBS solution for 24 hours. **(d)** A confocal fluorescence image with projected images of the hybrid prototissues at 24 hours. The gray, green and red regions indicated the non-labeled GUVs at the bottom layer, green GUVs at the edge of top layer, and the cells in the middle of the top layer. The scale bars were 100 μm .

7. Line 237 – typography error – *GOx-GVUs* instead of *GOx-GUVs*

Thank the reviewer for the comment. We have corrected “GOx-GVUs” to “GOx-GUVs” in page 11.

8. Line 259 – *What happens when the prototissues are detached without CaCl₂ washing? How long do the prototissues survive once detached from their production grids?*

Thank the reviewer for the comments. The prototissues disassembled into dispersed individual GUVs and amorphous GUVs blocks when they were detached from nickel mesh without CaCl₂ treatment. The prototissues can survive for 8 days after they were detached from the nickel mesh.

We added the below sentence in page 12 of the manuscript.

Page 12, ‘The prototissues were disassembled into dispersed individual GUVs and amorphous GUVs blocks when they were detached from nickel mesh without CaCl₂ treatment (Supplementary Fig. 13).’.

Supplementary Fig. 13 The fluorescence image of the prototissues composed of green GUVs detached from the NM grids without CaCl₂ treatment. The scale bar was 100 μm .

Page 12, ‘The detached prototissues survived for 8 days (Supplementary Fig. 12a). There were almost no GUVs dissociating from the prototissues by measuring the fluorescence of the solution in the petri-dish within 8 days (Supplementary Fig. 12b, c and d).’.

Supplementary Fig. 12 Stability of the detached prototissues. (a) The fluorescence images of a prototissue composed of green and red (1:1) GUVs at different days. The scale bars were 100 μm . (b) Fluorescence intensity of the mixed GUVs solution containing NBD-PE labelled GUVs ($1 \times 10^6/\text{mL}$) and Texas red-DHPE labelled GUVs ($1 \times 10^6/\text{mL}$). (c) The calibration curves of the GUVs solutions against concentration. a_0 was the concentration of $1 \times 10^6/\text{mL}$ NBD-PE labelled GUVs or $1 \times 10^6/\text{mL}$ Texas red-DHPE labelled GUVs. (d) Fluorescence intensity of the solution in the petri-dish containing detached same batch prototissues of (a) as a function of time.

9. Line 261 – How do you know that molecules larger than 20kDa could not penetrate the prototissue? Did you test the tissue permeability of larger fluorescent dextrans? If you did not please amend the manuscript to state that the upper permeability limit has not been measured.

Thank the reviewer for the comments. We did not test the prototissue permeability with larger fluorescent dextrans. We modified the sentence in page 12 of the manuscript as below.

Page 12, ‘Although the upper permeability limit was not measured, the molecules smaller than 20 kDa were allowed to enter the interior of prototissues (Fig. 5b).’.

10. Fig 5 – why did you use a hydroxyurea + HRP approach to generating NO here instead of using the arginine system in figure 4? It would be good to discuss this change

of strategy in the manuscript – what is the advantage of encapsulating HRP over arginine in prototissues?

Thank the reviewer for the comments. We added below sentences in page 12.

Page 12, ‘The advantage of prototissues containing GOx-GUVs and HRP-GUVs is that NO can be produced continuously in the presence of glucose and hydroxyurea. While the prototissues containing GOx-GUVs and Arginine-GUVs only produce limited NO due to the amount of arginine inside prototissues.’.

11. Line 293 – HEPEs? Should be HEPES?

Thank the reviewer for the comment. We have corrected the ‘HEPEs’ to ‘HEPES’ in page 14.

12. For vasodilation experiments, the concentration of prototissues is listed as 1×10^4 ? How did you calculate this tissue concentration?

Thank the reviewer for the comments. The square nickel meshes containing 2116 grids were used to prepare prototissues. The number of integral prototissues detached from the nickel mesh and the number of grids were counted under the same microscope views (n=15) to obtain the successful rate of 91.6%. Therefore, the number of prototissues from each nickel mesh was estimated to be about 1938. Five nickel meshes were used to prepare the prototissues at the same time to obtain about 1×10^4 prototissues. We added the below sentences in page 17 of the manuscript.

Page 17, ‘The square nickel meshes containing 2116 grids were used to prepare NO-prototissues. Each nickel meshes produced about 1938 NO-prototissues.’.

Page 17, ‘When the tension force reached equilibrium, about 1×10^4 NO-prototissues detached from the 5 NMs composed of GOx-GUVs with melittin pores and HRP-GUVs and 10 mM hydroxyurea were added into the bath solution.’.

13. In fig 6, how close were the prototissues to the aorta? Could you provide images of the experimental setup, or additional schematics that show the distance between aorta and prototissue? It is hard to visualise from the cartoon schematic.

Thank the reviewer for the comments. The schematic illustration of the vascular ring experimental setup was provided in Supplementary Fig. 14. Since the prototissues sank at the bottom of the bath, the distance between the bath bottom and the vascular ring was the distance between the vascular ring and the prototissues, which is 6 mm. We added the below sentences in page 14 of the manuscript.

Page 14, ‘The schematic illustration of the vascular ring experimental setup was provided in Supplementary Fig. 14’.

Supplementary Fig. 14 The schematic diagram of the prototissue-induced vasodilation in vitro. The aortic ring was hung on the fixed hooks connected to a force transducer in an organ bath containing 10 mL HEPES solution under physiological O₂ conditions at 37 °C. The prototissues located at the bottom of the bath, which was about 6 mm perpendicularly below the aorta ring.

14. Fig 6 – spelling errors – relaxing, hadroxyurea

Thank the reviewer for the comments. We have corrected the ‘relaxating’, ‘hadroxyurea’ to ‘relaxing’, ‘Hydroxyurea’ in Figure 6.

15. In your conclusion you briefly mention the use of prototissues in transplantation applications. Could you briefly expand on how prototissues could be used in this or other biomedical applications? Are there other potential applications for the prototissues generated here?

Thank the reviewer for the comments. There are still many challenges to transplant the NO-prototissues in vivo. The NO-prototissues may have potential in cancer treatment etc. Therefore, we modified the last sentence in conclusion as below.

‘These prototissues were confirmed to be capable of relaxing the rat blood vessels, consequently to improve their functions, which may hold potential to treat cardiovascular diseases.’.

Reviewer #2 (Remarks to the Author):

The work of Prof Han and coworkers describe a spatially organized system of prototissues, assembled from two or more type of giant unilamellar vesicles (GUVs). The assembly is succeeded in a paramagnetic solution and with the assistance of an external magnet and a nickel mesh placed underneath the assembly of the mesh. They also nicely demonstrate two-step and three-step signal communications in the prototissues using cascade enzyme reactions. In such a demonstration they produce NO and use this for biological response in a blood vessel tissue. The paper is well organized and clearly written. The claims are supported by the data provided.

The group has already shown in a previous paper published in Nat Comm ref. 13 (<https://doi.org/10.1038/s41467-019-14141-x>) that they can nicely assemble in 3D the GUVs. Based on that I see the advance in this paper is mostly on the demonstrations where they used the assemblies for the cascade reactions and so on. These demonstrations are good and convincing, however, I don't see why and how the prototissues achieved here will be beneficial for the advance of the field. What I mean here; is do these prototissues have to be well programmed and submillimeter sized to be able to function as in these demos. Because authors perform their experiments in Fig 5 and Fig 6 in solutions and these experiments will indeed give similar results even if the authors would use a pellet of bulk assemblies of the same GUVs, e.g. that is obtained via sedimentation of the same types of GUVs sequentially to obtain an assembly of the GUVs on top of each other. The possibility of having sub-millimeter programmed assemblies of GUVs does not seem to me the only way to obtain the reactions of the blood vessel or the cascade reactions shown in the paper. These reasons make me wonder if this follow up paper has built enough advance on the previous work to qualify being published at Nature Comm.

Thank the reviewer for the comments. The advancements of this paper over our previous one (*Nat. Commun.* 2020, **11**, 232) are as follows:

1. The detached prototissues capable of NO-generation were confirmed to be able to relax the living blood vessel, which was a step forward towards the biomedical application. This point was recognized by reviewer 1 "A key difference between this manuscript and previous work is connecting novel and relevant external inputs (e.g. hydroxyurea) to generate new reaction sequences in the prototissue - ultimately producing nitric oxide (NO) – and connecting this to a biological system in order to control its behaviour."
2. The high throughput spatial coded and detachable prototissues were generated using a simple, cost effective method, which was a critical step for their further applications.
3. The complex signal communication was achieved in the programmed ternary hybrid prototissues, which provided a way to fabricate more complicated prototissues with high-order functions.

In nature, there exist sub-millimeter sized and spatial coded tissues, such as pancreatic

islets (*Sci. Adv.* **5**, eaax4520, 2019; *Nat. Rev. Drug Discov.* **20**, 920-940, 2021). Pancreatic islets are with 50-400 μm in diameter (*Biosensors & Bioelectronics*, **24**, 113215, 2021), with most α cells at the peripheral region and β cells at inner region. We are on the way to fabricate multiple cell tissues with spatial organizations to mimic these tissues using this technique.

Pancreatic islets.
Sci. Adv. **5**, eaax4520 (2019)

Schematic of pancreatic islet.
Nat. Rev. Drug Discov. **20**, 920-940 (2021)

There is a typo in line 299, it is written 'molecure' instead of molecule.

Thank the reviewer for the comment. We have corrected the 'molecures' to 'molecules' in page 14 of the manuscript.

Reviewer #3 (Remarks to the Author):

This manuscript presents an interesting novel technology, built upon recent advancements in engineering synthetic cells and tissues.

The writing is mostly clear, results are presented and analyzed in convincing and conclusive way.

Introduction:

1. In the first paragraph of the results and discussion section, the magneto-Archimedes effect and the composition of the prototissues are explained, however it would be helpful to mention this early in the introduction as these are key to understanding the background of the paper. The rest of the introduction is successful in orienting the reader to what will be described in the paper.

Thank the reviewer for the comment. We added below sentences in page 2.

Page 2, ‘Diamagnetic materials move to the weak magnetic field area in a nonhomogeneous magnetic field. This phenomenon is called magneto-Archimedes effect. GUVs and living cells are diamagnetic materials, which were used as the building blocks for prototissues¹³.’.

Results and Discussion:

2. A minor comment, but when describing the magnetic field regions on the nickel mesh (NM) structures, it is mentioned that the weak magnetic field regions are the blue areas in Fig 1b – there are multiple blue areas and it would be helpful to specify that the weak regions are the light and/or the dark blue regions. However, the rest of the figures are useful in visualizing the results and have clear explanations.

Thank the reviewer for the comments. We have corrected the ‘blue’ to ‘dark blue’ in page 3 and 5 of the manuscript.

Page 3, ‘The NM exhibits a strong magnetic response and causes a gradient magnetic field inside NM (Fig. 1b, bottom layer) and 140 μm above NM (Fig. 1b, top layer), where the dark blue areas indicate weak magnetic field regions.’, ‘As expected, the GUVs were firstly assembled in the weak magnetic field regions (dark blue areas in Fig. 1b, bottom layer) inside each grid (Fig. 1c I). With the number of added GUVs increasing, GUVs gradually filled the space inside each grid of the NM to generate 60 μm thick bottom layer (Fig. 1c II) and further protruded to form 140 μm thick brick-shape top layer above the GUVs aggregations inside each grid (Fig. 1c III, IV), which were consisted with simulated field distribution (dark blue areas in Fig. 1b, top layer).’.

Page 5, ‘Dark blue areas indicated the weak magnetic field regions.’.

3. In the section “Diverse multi-compartment prototissues”, it is mentioned that assembling the multi-component prototissues can be attained by adjusting the distribution of the magnetic fields. Including a reference to the top box of Figure 2 here would help explain the differences in these fields. As described in this section, the level of control in the formation and structure of the prototissues is impressive.

Thank the reviewer for the comments. We added a reference to the top box of Figure 2 to explain the difference of the fields.

Fig. 2 Diverse multi-component prototissues. **a** Purple arrow, yellow arrow and cyan line indicated the vertical, inclined and no magnetic field, respectively. Red, green and gray rings indicated the red, green and non-labeled GUVs, respectively. **b** Schematic and fluorescence images of a GUVs prototissue of ‘eye’ structures with rGUVs inside gGUVs under a vertical magnetic field with the addition of gGUVs ($3 \times 10^5/\text{mL}$) and rGUVs successively ($2 \times 10^5/\text{mL}$). **c** Schematic and fluorescence images of a GUVs prototissues of modified ‘eye’ structures by trapping successively gGUVs ($3 \times 10^5/\text{mL}$) under a vertical magnetic field and rGUVs ($1 \times 10^5/\text{mL}$) under an inclined magnetic field. **d** Schematic and fluorescence

images of prototissues by trapping successively gGUVs ($1 \times 10^5/\text{mL}$) and rGUVs ($1 \times 10^5/\text{mL}$) under an inclined magnetic field. **e** Schematic and fluorescence images of binary prototissues of 'protruded structures' by trapping the mixture of gGUVs ($6 \times 10^5/\text{mL}$) and rGUVs ($6 \times 10^5/\text{mL}$) under a vertical magnetic field. **f** Schematic and fluorescence images of prototissues of 'protruded structures' with gGUVs at the bottom and rGUVs at the top by trapping successively gGUVs ($6 \times 10^5/\text{mL}$) and rGUVs ($4 \times 10^5/\text{mL}$) under vertical magnetic field. **g** Schematic and fluorescence images of prototissues with two layered structures by trapping successively gGUVs ($1.2 \times 10^6/\text{mL}$) under a vertical magnetic field and rGUVs ($4 \times 10^5/\text{mL}$) in the absence of magnetic field. **h** Schematic and fluorescence images of prototissues composed of three components by trapping successively non-labeled GUVs ($6 \times 10^5/\text{mL}$) and gGUVs ($6 \times 10^5/\text{mL}$) under vertical magnetic field to form 'protruded structures', and subsequently rGUVs ($2 \times 10^5/\text{mL}$) in the absence of magnetic field. All the prototissues were assembled on the top of a circular magnet with 0.3 T magnetic flux density. After one type of GUVs were trapped, the time intervals were 1 hour before adding another type of GUVs. The scale bars were 100 μm .

4. I wonder if an interesting addition to the multi-component prototissue communication assays would be including an experiment where the communication GUVs (i.e. Gox-GUVs) are free-floating in buffer to understand how they may communicate less/more effectively when in solution than when in the prototissue structure.

Thank the reviewer for the comments. The signal communication between free-floating GOx-GUVs ($1.2 \times 10^6/\text{mL}$) and HRP-GUVs ($4 \times 10^5/\text{mL}$) was investigated. Glucose molecules (30 mM) and Amplex Red (0.05 μM) were added into the solution to initiate the reaction. These experimental conditions were the same with those for the communication assay between GOx-GUVs and HRP-GUVs populations in the binary prototissues. The fluorescence intensity of resorufin molecule (585 nm, Supplementary Fig.9 a) in the solution was monitored as a function of time (Supplementary Fig.9 b). The maximum intensity was observed at about 60 minutes, which was about 3 times of that from the GOx-GUVs/HRP-GUVs binary prototissues. It implied the close packing of GUVs in the prototissues promoted chemical communications.

We added the below sentences in page 9 of the manuscript.

Page 9, 'As the reactants entered into the GUVs continuously, the red fluorescence gradually became stronger and reached the equilibrium in the non-labeled GUVs regions at about 20 minutes (Fig. 3d, e), which was 3 times faster than that from the floating GOx-GUVs and HRP-GUVs at the same experimental conditions (Supplementary Fig. 9). It implied the close packing of GUVs in the prototissues promoted chemical communications.'

Supplementary Fig. 9 (a) The emission spectrum of resorufin. **(b)** The fluorescence intensity of resorufin in the solution containing free floating GOx-GUVs ($1.2 \times 10^6/\text{mL}$) and HRP-GUVs ($4 \times 10^5/\text{mL}$) as a function of time after the addition of glucose molecules (30 mM) and Amplex Red ($0.05 \mu\text{M}$) simultaneously (red curve). The fluorescence intensity of Amplex Red ($0.05 \mu\text{M}$) in the solution without GOx-GUVs, HRP-GUVs, and glucose as a function of time (blue curve).

5. In the section “Prototissues capable of producing NO,” was there a test to see if/how many GUVs are dissociating from the proto-tissue structure? Measuring the fluorescence of the buffer in the petri-dish over time would be a good way to measure if GUVs encapsulating these different components are dissociating from the prototissues.

Thank the reviewer for the comments. As suggested by the reviewer, we carried out the experiments to measure the dissociation of GUVs from the prototissues. Green-GUVs ($1 \times 10^6/\text{mL}$, labeled with NBD PE) and Red-GUVs ($1 \times 10^6/\text{mL}$, labeled with Texas red DHPE) were mixed to fabricate the prototissues. The fluorescence of NBD PE ($\lambda_{\text{ex}} = 463 \text{ nm}$, $\lambda_{\text{em}} = 539 \text{ nm}$) and Texas red DHPE ($\lambda_{\text{ex}} = 561 \text{ nm}$, $\lambda_{\text{em}} = 598 \text{ nm}$) were used to monitor the concentration of Green-GUVs and Red-GUVs in the solution respectively. The calibration curves of Green-GUVs and Red-GUVs were obtained (Supplementary Fig. 12b, c). No fluorescence of NBD PE and Texas red DHPE was observed in the solution containing prototissues within 8 days (Supplementary Fig. 12d), which implied almost no GUVs dissociation from the prototissues.

We added below sentences in page 12 in the manuscript.

Page 12, ‘There were almost no GUVs dissociating from the prototissues by measuring the fluorescence of the solution in the petri-dish within 8 days (Supplementary Fig. 12b, c and d).’

Supplementary Fig. 12 Stability of the detached prototissues. (a) The fluorescence images of a prototissue composed of green and red (1:1) GUVs at different days. The scale bars were 100 μm . (b) Fluorescence intensity of the mixed GUVs solution containing NBD-PE labelled GUVs ($1 \times 10^6/\text{mL}$) and Texas red-DHPE labelled GUVs ($1 \times 10^6/\text{mL}$). (c) The calibration curves of the GUVs solutions against concentration. a_0 was the concentration of $1 \times 10^6/\text{mL}$ NBD-PE labelled GUVs or $1 \times 10^6/\text{mL}$ Texas red-DHPE labelled GUVs. (d) Fluorescence intensity of the solution in the petri-dish containing detached same batch prototissues of (a) as a function of time.

6. It is very exciting to see that the prototissues can relax blood vessels by release of NO in the section “NO-prototissues for vasodilation.” It should be clarified here that this work was done *in vitro*.

Thank the reviewer for the comments. We added ‘*in vitro*’ in the subtitle of the section ‘**NO-prototissues for vasodilation *in vitro***’ in page 14. We also modified the sentence of ‘The thoracic aorta of rat was isolated from the body and cut into aorta rings with the length of 5 mm, which were subsequently mixed with NO-prototissues *in vitro* in the organ bath.’ in page 14.

7. The authors write that the NO-prototissues are “mixed” with living blood vessels, can they elaborate? What is the spatial relationship of the prototissues with the blood vessels—are they free floating in solution together?

Thank the reviewer for the comments. During the vasodilation experiments, the NO-prototissues was added into the organ bath perpendicular below the vascular ring on the

hooks connected to a force transducer (Supplementary Fig. 14). Since the prototissues sank at the bottom of the bath, the distance between the bath bottom and the vascular ring was the distance between the vascular ring and the prototissues, which was 6 mm. We added the below sentences in page 14 of the manuscript.

Page 14, 'The schematic illustration of the vascular ring experimental setup was provided in Supplementary Fig. 14'.

Supplementary Fig. 14 The schematic diagram of the prototissue-induced vasodilation in vitro. The aortic ring was hung on the fixed hooks connected to a force transducer in an organ bath containing 10 mL HEPES solution under physiological O₂ conditions at 37 °C. The prototissues located at the bottom of the bath, which was about 6 mm perpendicularly below the aorta ring.

8. In Figure 6, it is written that the blood vessels were treated using NO-prototissues and NO-free-prototissues. Including images of the blood vessels treated by pure NO in solution (not GUVs) would be more rigorous.

Thank the reviewer for the comments. Sodium nitroprusside (SNP) as the common NO donor was used to treat the blood vessel (*Circulation* **95**, 2303-2311, 1997; *Nitric Oxide* **69**, 56-60, 2017). The experimental results and corresponding descriptions were added in page 14 as below.

Page 14, 'On the contrary, sodium nitroprusside (SNP) as the common NO donor was used to treat aorta ring^{30,31}. A rapid decrease of tension force was observed immediately upon the addition of 0.1 μM SNP at 20 minutes (Fig. 6d).'

'With the treatment of 0.1 μM SNP, the tension force decreased by 2.84 g with the blood vessel relaxation of 43.3% (Fig. 6e, f, blue columns).'

Fig 6. NO-prototissues for vasodilation in vitro. **a** Schematic illustration of vasodilation induced by NO from NO-prototissues. Representative tension curve of vasodilation against time in the presence of NO-prototissues composed with GOx-GUVs and HRP-GUVs (**b**), NO-free-prototissues composed of GOx-free GUVs and HRP-GUVs (**c**) and **sodium nitroprusside (SNP)** (**d**). The black arrow indicated the point of the addition of hydroxyurea (Ha, 10 mM) or **SNP** (0.1 μ M). **e, f** Bar charts of the decrease in tension force and relaxation of vascular rings observed in the presence of NO-prototissues (2.59 ± 0.77 g; $31.2 \pm 10.1\%$ relaxation, **black columns**), NO-free-prototissues (0.14 ± 0.12 g; $3.4 \pm 2.9\%$ relaxation, **red columns**), and 0.1 μ M **SNP** (2.84 ± 0.92 g; $43.3 \pm 9.0\%$ relaxation, **blue columns**), $n=5$. **g** Fluorescence images of vascular sections stained with DAF-FM DA (the green fluorescence channel responded to NO production, left column), DAPI (blue fluorescence with nuclei, middle column), and their merge images (right column). The blood vessels were treated using NO-prototissues (top row) or NO-free-prototissues (middle row) in the HEPES solution containing 10 mM hydroxyurea for 20 minutes, respectively. The blood vessel was treated using 0.1 μ M **SNP** in the HEPES solution for 20 minutes (bottom row). The scale bars were 100 μ m.

30. Ito, N., Bartunek, J., Spitzer, K. W. & Lorell, B. H. Effects of the Nitric Oxide Donor Sodium Nitroprusside on Intracellular pH and Contraction in Hypertrophied Myocytes. *Circulation* **95**, 2303-2311 (1997).

31. Orfanidou, M. A., Lafioniatis, A., Trevlopoulou, A., Touzlatzi, N. & Pitsikas, N. Acute and repeated exposure with the nitric oxide (NO) donor sodium nitroprusside (SNP) differentially modulate responses in a rat model of anxiety. *Nitric Oxide* **69**, 56-60 (2017).

9. Finally, in the summary, the authors write that these prototissues could be used to treat cardiovascular diseases through transplantations. I think this is a thrilling and plausible speculation based on the work described in this paper. Can the authors speculate on how the prototissues may be bound to distinct areas in the body, and how long would these GUVs be able to relax the vessels?

Thank the reviewer for the comments. The NO-prototissues may be bound to the greater omentum in the body. However, there are still many challenges to transplant these

prototissues in vivo. Therefore, we modified the last sentence in conclusion as below.

‘These prototissues were confirmed to be capable of relaxing the rat blood vessels, consequently to improve their functions, **which may hold potential to treat cardiovascular diseases.**’.

Methods:

10. Are the prototissues on and off the NM stored in buffer in the petri-dish? I think this would be important to include in the results section, as well as methods. It should again be clarified in the methods that vasodilation stimulated by NO from NO-tissues was done in vitro.

Thank the reviewer for the comments. We made below modification in the section of results and discussion, and the section of methods.

The words ‘**on the NM**’ were added in page 10 and 15.

The sentence ‘**The prototissues were on the NM during the signal communications.**’ was added in page 17.

The words ‘**detached from NM**’ were added in page 14, 15 and 17.

The words ‘in vitro’ were added in the subtitle of the section ‘**NO-prototissues for vasodilation in vitro**’ in page 17.

REVIEWERS' COMMENTS

Reviewer #1 (Remarks to the Author):

I was pleased to read the revised manuscript by Han and coworkers which contains new experiments including quantification of prototissue stability, response to osmotic stress and cell viability within hybrid prototissues. This has improved an already excellent manuscript, and increased the clarity around the fabrication and material properties of the tissues. The authors have largely addressed all of my previous comments by making these changes, but I have one minor comment regarding the viability of glioma cells (Supplementary Figure 11).

Significant cell death is observed over 12 hours, which is the same rate as when the cells are suspended in PBS. I agree with the authors conclusion that the prototissue does not negatively affect the cell viability (I assume cell death is due to long term incubation in PBS), but it would be interesting for the authors to compare these results against their previously published work (<https://doi.org/10.1038/s41467-019-14141-x>), where hybrid prototissues containing cells are formed. In this paper, the cells appeared to stay viable over a 6 hour experimental timeframe. Could the authors comment on the difference in cell viability in these prototissues in the two manuscripts? Making such a comparison would be interesting for readers interested in the exciting translational potential of these prototissue materials.

Reviewer #3 (Remarks to the Author):

The Authors answered all my comments and questions. Thank you!

Response to the reviewers' comments

For the sake of clarity, the comments of the reviewer have been collated in black, and our response to each comment appears in blue. All the changes to the manuscript are highlighted in red.

Reviewer #1 (Remarks to the Author):

I was pleased to read the revised manuscript by Han and coworkers which contains new experiments including quantification of prototissue stability, response to osmotic stress and cell viability within hybrid prototissues. This has improved an already excellent manuscript, and increased the clarity around the fabrication and material properties of the tissues. The authors have largely addressed all of my previous comments by making these changes, but I have one minor comment regarding the viability of glioma cells (Supplementary Figure 11).

Thank the reviewer for the careful review and the positive recognition of our work.

Significant cell death is observed over 12 hours, which is the same rate as when the cells are suspended in PBS. I agree with the authors conclusion that the prototissue does not negatively affect the cell viability (I assume cell death is due to long term incubation in PBS), but it would be interesting for the authors to compare these results against their previously published work (<https://doi.org/10.1038/s41467-019-14141-x>), where hybrid prototissues containing cells are formed. In this paper, the cells appeared to stay viable over a 6 hour experimental timeframe. Could the authors comment on the difference in cell viability in these prototissues in the two manuscripts? Making such a comparison would be interesting for readers interested in the exciting translational potential of these prototissue materials.

Thank the reviewer for the comment. In the previous paper (*Nat. Commun.* **11**, 232 (2020), <https://doi.org/10.1038/s41467-019-14141-x>), the hybrid prototissues composed of giant unilamellar vesicles (GUVs) encapsulating glucose oxidase and cells were fabricated to study the chemical communication between GUVs and cells. In 400 mM glucose solution, the glucose molecules entered the GUVs, consequently to be catalyzed by glucose oxidase (GOD) to generate H_2O_2 . H_2O_2 diffused into cells and killed them (see below image from *Nat. Commun.* **11**, 232 (2020)). Therefore, almost all cells died after 6 hours. In another words, the cells were killed deliberately in the previous work. In the current manuscript, the cells were incubated only in PBS solution. Therefore, the cells survived more longer. We added the below sentences in page 8 of the manuscript.

(*Nat. Commun.* **11**, 232 (2020))

‘The glioma cells in the hybrid prototissues in the PBS solution survived longer than the cells in the prototissues composed of GUVs (capable of producing H₂O₂ in the presence of glucose) and cells in the glucose solution¹³, which was due to the toxicity of H₂O₂ generated by GUVs.’.

Reviewer #3 (Remarks to the Author):

The Authors answered all my comments and questions. Thank you!

Thank the reviewer for the review, comments and recognition of our work.